# Two-Stage Limited-Information Estimation for Structural Equation Models of Round-Robin Variables

**Terrence D. Jorgensen** [1,*] , **Aditi M. Bhangale** [1] **and Yves Rosseel** [2]

1 Research Institute of Child Development and Education, University of Amsterdam, Nieuwe Achtergracht 127, 1018 WS Amsterdam, The Netherlands

2 Center for Data Analysis and Statistical Science, Ghent University, Krijgslaan 281-S9, 9000 Gent, Belgium; yves.rosseel@ugent.be

* Correspondence: t.d.jorgensen@uva.nl

**Abstract:** We propose and demonstrate a new two-stage maximum likelihood estimator for parameters of a social relations structural equation model (SR-SEM) using estimated summary statistics ($\widehat{\Sigma}$) as data, as well as uncertainty about $\widehat{\Sigma}$ to obtain robust inferential statistics. The SR-SEM is a generalization of a traditional SEM for round-robin data, which have a dyadic network structure (i.e., each group member responds to or interacts with each other member). Our two-stage estimator is developed using similar logic as previous two-stage estimators for SEM, developed for application to multilevel data and multiple imputations of missing data. We demonstrate out estimator on a publicly available data set from a 2018 publication about social mimicry. We employ Markov chain Monte Carlo estimation of $\widehat{\Sigma}$ in Stage 1, implemented using the R package `rstan`. In Stage 2, the posterior mean estimates of $\widehat{\Sigma}$ are used as input data to estimate SEM parameters with the R package `lavaan`. The posterior covariance matrix of estimated $\widehat{\Sigma}$ is also calculated so that `lavaan` can use it to calculate robust standard errors and test statistics. Results are compared to full-information maximum likelihood (FIML) estimation of SR-SEM parameters using the R package `srm`. We discuss how differences between estimators highlight the need for future research to establish best practices under realistic conditions (e.g., how to specify empirical Bayes priors in Stage 1), as well as extensions that would make 2-stage estimation particularly advantageous over single-stage FIML.

**Keywords:** structural equation model; social relations model; social network data; round-robin design; maximum likelihood estimation; two-stage estimation; Markov chain Monte Carlo estimation

## 1. Introduction

Two-stage estimation is frequently applied in a variety of latent variable models. For example, a structural equation model (SEM) has both measurement and structural components (see Section 1.2 for details), and there is a long history of first using factor analysis to estimate factor scores and subsequently treating the latter as data in a path analysis (see [1] for an overview of such methods). The same approach is taken in item response theory (IRT) by estimating plausible values of person parameters [2]. Individual scores on categorical latent variables (i.e., latent class membership) can also be estimated to use in subsequent structural analyses [3,4]. Two-stage estimation can be particularly advantageous when estimating latent components of complexly structured variables, such as multilevel data [5,6] and network data [7]. In this paper, we propose a two-stage algorithm to estimate parameters of a social relations SEM (SR-SEM; [8,9]) for multivariate dyadic network data. Our algorithm is inspired by similar ones proposed for estimating multilevel SEM (ML-SEM) parameters [6] and for efficiently estimating SEM parameters with multiply imputed data [10,11], both of which involve limited information analysis of summary statistics (i.e., covariance matrices).

Our paper is structured as follows. Section 1.1 introduces dyadic network data and the social relations model (SRM), including an overview of extensions and estimators in

Section 1.1.2. Section 1.1.3 introduces multiple imputation, which is relevant for understanding how our Stage 1 results are prepared for Stage 2 estimation. We briefly introduce SEM in Section 1.2, with some discussion about two-stage estimation approaches for multiply imputed data in Section 1.2.1 and for ML-SEM in Section 1.2.2. We then introduce the SR-SEM in Section 1.2.3, which Nestler et al. [8,9] first proposed using single-stage maximum likelihood estimation (MLE)—we refer to this as their full-information (FIML) estimator. We describe our proposed two-stage MLE algorithm in Section 2 in the context of an application to empirical data made publicly available on the Open Science Framework (OSF; [12]) by Salazar Kämpf et al. [13]. Results of two-stage MLE and FIML are compared in Section 3, with software syntax provided in the appendices and on our OSF project (https://osf.io/2qs5w/). Section 4 discusses advantages and potential extensions of the two-stage estimator, as well as necessary future research to validate the proposed method.

### 1.1. Social Relations Modeling

Dyadic variables are measured once for each member of a pair (e.g., friends, work colleagues, romantic couples), and dyadic network data occur when each member in a group provides data about each other member in the group (e.g., how much they like each other person). Such data have a complex nesting structure, such that a bivariate response vector $y_{\{ij\}}$ (e.g., person $i$'s liking of person $j$ and vice versa) is dependent upon outgoing and incoming random effects that are correlated within individuals. The SRM [14] is a dyadic network model developed for (approximately) continuous data (e.g., random effects and residuals are assumed to follow a Gaussian or "normal" distribution), and it is a special case of Hoff's [15,16] additive and multiplicative effects model for network data (AMEN). Other related models include the latent interdependence model [17] and models for traditional (i.e., dichotomous) social network data that indicate the presence/absence of ties: $p_2$ [18,19] and $j_2$ [20]. We focus only on the SRM in this paper because other modeling frameworks for social network data—such as the stochastic actor-oriented model [21] and exponential random graph (or $p^*$) model [22–24]—deviate further from the SRM's focus on individuals and dyads.

The univariate SRM can be depicted as a random effects model [25,26] for the dyadic vector $\mathbf{y}_{g\{ij\}}$, where the braces indicate that the ordering of members $i \neq j$ in group $g \in 1, \ldots, G$ is arbitrary. Because each case $i \neq j \in 1, \ldots, N_g$ in group $g$ belongs to multiple dyads $d_g \in 1, \ldots, D_g$, each bivariate dyadic observation is nested in the set of observations in which case $i$ is a member as well as in the set of observations in which case $j$ is a member. The SRM decomposes $\mathbf{y}_{g\{ij\}}$ into case- and dyad-level components:

$$\mathbf{y}_{g\{ij\}} = \begin{bmatrix} y_{gij} \\ y_{gji} \end{bmatrix} = \begin{bmatrix} \mu_g + E_{gi} + A_{gj} + R_{gij} \\ \mu_g + E_{gj} + A_{gi} + R_{gji} \end{bmatrix}, \tag{1}$$

where $\mu_g$ is the expected value of the observations (e.g., average amount of liking) in group $g$. The subscript $g$ can be dropped when modeling a single round-robin group/network, or $\mu$ can be dropped altogether when data are modeled as (group-)mean centered. $E_{gi}$ and $A_{gj}$ are case-level ego (outgoing) and alter (incoming) effects, respectively—for example, $E_{gi}$ would represent how much person $i$ likes others in general, and $A_{gj}$ would represent how much person $j$ is generally liked by others (i.e., likability). Each $R$ is a dyad-level residual, which captures any relationship-specific effects (e.g., how much $i$ uniquely likes $j$ beyond what is expected from their case-level effects). More descriptive terms have been used for $E_{gi}$ and $A_{gj}$, such as actor and partner effects when $\mathbf{y}_{g\{ij\}}$ are behavioral interactions (e.g., social mimicry; [13]) or perceiver and target effects when $\mathbf{y}_{g\{ij\}}$ are interpersonal perceptions (e.g., of personality traits; [27]). However, case-level units of analysis might not be people—the SRM can be applied to networks of countries, communities, or households [28].

Each case's vector of ego and alter effects is stored in a vector $\mathbf{u}_{gi}$, assumed bivariate normally distributed:

$$\mathbf{u}_{gi} = \begin{bmatrix} E_{gi} \\ A_{gi} \end{bmatrix} \sim \mathcal{N}\left( \begin{bmatrix} 0 \\ 0 \end{bmatrix}, \Sigma_{EA} = \begin{bmatrix} \sigma_E^2 & \\ \sigma_{EA} & \sigma_A^2 \end{bmatrix} \right), \tag{2}$$

where larger case-level variances $\sigma_E^2$ and $\sigma_A^2$ indicate greater heterogeneity of outgoing or incoming effects, respectively. The correlation $\rho_{EA} = \frac{\sigma_{EA}}{\sigma_E \sigma_A}$ is referred to as *generalized reciprocity* [27], which would be positive if (for example) those who have a propensity to like people (less) are also generally (un)likable.

Each dyad's pair of residuals is stored in a vector $\mathbf{r}_{g\{ij\}}$, also assumed bivariate normally distributed:

$$\mathbf{r}_{g\{ij\}} = \begin{bmatrix} R_{gij} \\ R_{gji} \end{bmatrix} \sim \mathcal{N}\left( \begin{bmatrix} 0 \\ 0 \end{bmatrix}, \Sigma_R = \sigma_R^2 \begin{bmatrix} 1 & \\ \rho_R & 1 \end{bmatrix} \right), \tag{3}$$

where the variance $\sigma_R^2$ is assumed equal for indistinguishable dyads (see ch. 8 of [29]). Larger relationship variance indicates a greater degree to which (for example) relationship-specific liking differs from what would be expected given person $i$'s general tendency to like others and person $j$'s general tendency to be liked by others. The (residual) correlation $\rho_R$ between relationship effects is referred to as *dyadic reciprocity* [27], which would be positive if person $i$ particularly likes person $j$ (i.e., more than would be expected given person $i$'s general propensity for liking and person $j$'s general likability) while person $j$ also particularly likes person $i$ (i.e., the liking is mutual).

The means in Equation (1) can be considered fixed effects (which is preferable when modeling few round-robin groups) or random effects. The latter would entail assuming they follow a normal distribution,

$$\mu_g \sim \mathcal{N}(\mu, \sigma_G), \tag{4}$$

with grand mean $\mu$ and standard deviation $\sigma_G$. All (co)variances are generally assumed to be invariant across round-robin groups.

### 1.1.1. Multivariate SRM

Bivariate SRM data have long been analyzed using ANOVA decomposition, but full and restricted MLE were more recently proposed for multivariate SRM with several round-robin variables [30]. When they are indicators of the same construct, modeling multiple round-robin variables can allow for SRM effects to be disentangled from measurement error [25,27]. When each round-robin variable represents a different construct, richer research questions can be answered about correlations of SRM effects between variables. We present a two-variable example based on the Salazar Kämpf et al. [13] data on social mimicry and liking, about which we provide details in Section 2:

$$\begin{bmatrix} y_{1,gij} \\ y_{1,gji} \\ y_{2,gij} \\ y_{2,gji} \end{bmatrix} = \begin{bmatrix} \mu_{1,g} \\ \mu_{1,g} \\ \mu_{2,g} \\ \mu_{2,g} \end{bmatrix} + \begin{bmatrix} E_{1,gi} \\ A_{1,gi} \\ E_{2,gi} \\ A_{2,gi} \end{bmatrix} + \begin{bmatrix} A_{1,gj} \\ E_{1,gj} \\ A_{2,gj} \\ E_{2,gj} \end{bmatrix} + \begin{bmatrix} R_{1,gij} \\ R_{1,gji} \\ R_{2,gij} \\ R_{2,gji} \end{bmatrix}, \tag{5}$$

where group means for each variable remain equal within each dyad (see Equation (1)).

On average, there may be more liking within some round-robin groups than within other groups; likewise, average mimicry may vary across groups. These group-level effects

could be correlated, such that groups that exhibit more social mimicry tend to experience more liking. This is captured by the covariance $\sigma_{21,G}$ between these variables:

$$\begin{bmatrix} \mu_{1,g} \\ \mu_{2,g} \end{bmatrix} \sim \mathcal{N} \left( \mu_G = \begin{bmatrix} \mu_1 \\ \mu_2 \end{bmatrix}, \Sigma_G = \begin{bmatrix} \sigma^2_{1,G} & \\ \sigma_{21,G} & \sigma^2_{2,G} \end{bmatrix} \right). \tag{6}$$

Within each round-robin group, people who engage in more social mimicry with interaction partners (ego effect of Variable 1: $E_1$) might also tend to like others more than average (ego effect of Variable 2: $E_2$) or they might be liked more by others (alter effect of Variable 2: $A_2$). These associations would be captured by case-level covariances (or correlations, when standardized):

$$\mathbf{u}_{gi} = \begin{bmatrix} E_{1,gi} \\ A_{1,gi} \\ E_{2,gi} \\ A_{2,gi} \end{bmatrix} \sim \mathcal{N} \left( \begin{bmatrix} 0 \\ 0 \\ 0 \\ 0 \end{bmatrix}, \Sigma_{EA} = \begin{bmatrix} \sigma^2_{E_1} & & & \\ \sigma_{A_1,E_1} & \sigma^2_{A_1} & & \\ \sigma_{E_2,E_1} & \sigma_{E_2,A_1} & \sigma^2_{E_2} & \\ \sigma_{A_2,E_1} & \sigma_{A_2,A_1} & \sigma_{A_2,E_2} & \sigma^2_{A_2} \end{bmatrix} \right). \tag{7}$$

Group- and case-level (co)variances are unconstrained. However, analogous dyad-level correlations would require equality constraints [30], following similar logic as equal residual variances $\sigma_R$ in Equation (3): the order of members is arbitrary in an indistinguishable dyad. This yields *intra*personal and *inter*personal dyadic correlations between variables:

$$\mathbf{r}_{gi} = \begin{bmatrix} R_{1,gij} \\ R_{1,gji} \\ R_{2,gij} \\ R_{2,gji} \end{bmatrix} \sim \mathcal{N} \left( \begin{bmatrix} 0 \\ 0 \\ 0 \\ 0 \end{bmatrix}, \Sigma_R = \begin{bmatrix} \sigma^2_{R_1} & & & \\ \rho_{R_1}\sigma^2_{R_1} & \sigma^2_{R_1} & & \\ \sigma_{21,\text{intra}} & \sigma_{21,\text{inter}} & \sigma^2_{R_2} & \\ \sigma_{21,\text{inter}} & \sigma_{21,\text{intra}} & \rho_{R_2}\sigma^2_{R_2} & \sigma^2_{R_2} \end{bmatrix} \right). \tag{8}$$

For example, a positive intrapersonal correlation ($\rho_{21,\text{intra}} = \frac{\sigma_{21,\text{intra}}}{\sigma_{R_1}\sigma_{R_2}}$) would indicate that when person $i$ especially likes person $j$, person $i$ also especially mimics person $j$. Conversely, a positive interpersonal correlation ($\rho_{21,\text{inter}} = \frac{\sigma_{21,\text{inter}}}{\sigma_{R_1}\sigma_{R_2}}$) would indicate that when person $i$ especially mimics person $j$, person $j$ especially likes person $i$.

### 1.1.2. SRM with Covariate Effects

The univariate SRM in Equation (1) has been extended by adding covariates, both as predictors of random effects [7,28,31] and as auxiliary correlates [32]. Covariate effects can be of substantive interest, but they can also alleviate bias in parameter estimates by accounting for reasons why data went missing, thus justifying the missing-at-random (MAR) assumption made by modern missing-data methods [33].

When modeling multiple round-robin groups, the group-level means can be modeled as a function of group-level covariates ($\xi$):

$$\mu_g = \kappa_0 + \sum_{c=1}^{C} \kappa_c \xi_{g,c} + \zeta_g, \tag{9}$$

where $C$ is the number of group-level covariates, $\kappa_c$ is the effect of predictor $\xi_c$ on group means, and unexplained group differences are captured by the residuals $\zeta_g$. The intercept $\kappa_0$ will equal the grand mean only when all predictors are mean-centered. However, explaining group-level variance is rarely the focus of research employing round-robin designs. In fact, group-level variance is often negligible when people are randomly assigned to round-robin groups, as in our real-data example (Section 2.1).

When case-level covariates (**x**) predict ego and alter effects, the distributional assumption in Equation (2) applies to their residuals $\varepsilon$ and $\delta$:

$$\begin{bmatrix} E_{gi} \\ A_{gi} \end{bmatrix} = \begin{bmatrix} \sum_{p=1}^{P} \beta_p x_{gi,p} + \varepsilon_{gi} \\ \sum_{p=1}^{P} \alpha_p x_{gi,p} + \delta_{gi} \end{bmatrix}, \tag{10}$$

where $P$ is the number of case-level predictors, $\beta_p$ is the effect of predictor $x_p$ on ego effects, $\alpha_p$ is the effect of predictor $x_p$ on alter effects, and unexplained individual differences are captured by residuals $\varepsilon_{gi}$ and $\delta_{gi}$. For example, personality traits (**x**) such as openness to experience and extraversion could be used to predict general liking ($E$) and likability ($A$), respectively.

Likewise, dyad-level predictors $q = 1, \ldots, Q$ can be added to the Level 1 model:

$$\begin{bmatrix} y_{gij} \\ y_{gji} \end{bmatrix} = \begin{bmatrix} \mu_g + E_{gi} + A_{gj} + \sum_{q=1}^{Q} \gamma_q w_{gij,q} + \sum_{q=1}^{Q} \lambda_q w_{gji,q} + R_{gij} \\ \mu_g + E_{gj} + A_{gi} + \sum_{q=1}^{Q} \gamma_q w_{gji,q} + \sum_{q=1}^{Q} \lambda_q w_{gij,q} + R_{gji} \end{bmatrix}, \tag{11}$$

where the intercept $\mu_g$ can be substituted to include group-level covariates, as in Equation (9). The $E$ and $A$ terms in Equation (11) can also be substituted to incorporate predictors, as in Equation (10). Dyad-level predictors can vary within a dyad (i.e., $w_{gij} \neq w_{gji}$)—for example, how attractive or agreeable each person thinks the other person is. However, predictors could also be constant within a dyad ($w_{g\{ij\}} = w_{gij} = w_{gji}$)—for example, whether a dyad contains same- or other-sex members, or how long the dyad members have been acquainted. In the latter case, we cannot distinguish the intrapersonal from the interpersonal effects in Equation (11) ($\gamma_q = \lambda_q$), so the predictor $w_{\{ij\},q}$ should only be included once. However, even when a predictor $W$ varies within a dyad with indistinguishable members, intrapersonal ($\gamma$) and interpersonal ($\lambda$) effects are each constrained to equality across the bivariate observations [30], just as residual variances $\sigma_R$ are invariant in Equation (3).

Covariate effects have been estimated using MLE [26,34] and Markov chain Monte Carlo (MCMC) estimation [15,31,33], the latter of which treats SRM's random effects $[E_{gi}, A_{gi}]'$ as parameters to be estimated [31], similar to simpler hierarchical models with random effects [35]. This approach is called *data augmentation* [36] and can be applied not only to random effects but to other types of latent variable, such as factor scores in SEM [37], person parameters in IRT [38], and latent responses in probit regression [39]. Missing observations in partially observed variables can also be handled with data augmentation, as Jorgensen et al. [33] demonstrated for the univariate SRM with incomplete data.

Two-stage estimation techniques have also been proposed, which allow SRM components to enter models as predictors rather than being limited to outcomes as in Equation (10). Stage 1 estimates of SRM effects were traditionally obtained using least-squares methods [40] but can also be obtained from mixed models [13,25] or SEM [41]. Treating estimates of SRM effects as observed data yields an estimated covariate effect with underestimated uncertainty (i.e., *SE* too small, confidence interval [CI] too narrow). Lüdtke et al. [7] proposed using a modern missing-data technique to overcome this limitation: *multiple imputation*. We introduce multiple imputation in the next section, which also discusses how this method has been used for round-robin data.

### 1.1.3. Multiple Imputation of Plausible Values

To overcome limitations of single-imputation techniques, Rubin [42] proposed to replace ("impute") each missing value with a random sample of plausible estimates for what values could have been observed. This was developed from a Bayesian framework, following similar principles as data augmentation. However, rather than estimating parameters of the hypothesized analysis model, the goal is to save posterior samples of estimated missing values, which are the multiple imputations that "complete" multiple copies of the data set. Numerous algorithms are available to estimate distributions of missing values, described in tutorial articles [43,44] and books [45–47] that also discuss the assumptions

required for imputation models to yield unbiased estimates. Thus, rather than providing a full treatment of multiple imputation here, we cover only the details that are relevant to the scope of this article.

Analysis of multiple imputations can be roughly divided into three stages [45]. The first stage involves specifying a model for the sampling distribution of observed and missing values, from which imputed values are sampled. This can be specified jointly for all (or a subset of) incomplete variables (e.g., using a multivariate normal sampling distribution; [46]), or univariate imputation models can be specified for each incomplete variable (i.e., a "fully conditional specification" with "chained equations"; [47]). Imputation models can be relatively unrestricted or have constraints in line with hypothesized analysis models, the former reducing bias but the latter increasing precision [48]. Once data have been imputed $M$ times, Stage 2 involves fitting the hypothesized model(s) to each imputation—thus, this method falls within the diverse family of two-stage estimation techniques.

A final Stage 3 involves pooling the $M$ sets of results. Point and $SE$ estimates are pooled using "Rubin's rules" [42]. Pooled point estimates $\hat{\vartheta}$ are simply an average of all imputation-specific point estimates:

$$\hat{\vartheta} = \frac{1}{M} \sum_{m=1}^{M} \hat{\vartheta}_m. \tag{12}$$

Pooling $SE(\vartheta)$ requires summing two sources of sampling variance: a within-imputation component $W$ and a between-imputation component $B$,

$$W = \frac{1}{M} \sum_{m=1}^{M} SE(\vartheta_m)^2, \tag{13}$$

$$B = \frac{1}{M-1} \sum_{m=1}^{M} (\hat{\vartheta}_m - \hat{\vartheta})^2, \tag{14}$$

$$T = W + \left(1 + \frac{1}{M}\right) B, \tag{15}$$

where the total sampling variance $T$ is the squared $SE$ used for inference. The variance components quantify two sources of uncertainty about point estimates, arising from distinct aspects of sampling error. The first source of sampling variance ($W$) quantifies complete-data uncertainty, which arises from the random process of sampling from a population, thus introducing uncertainty even with complete data. The second source of sampling variance ($B$) quantifies missing-data uncertainty, which arises from the random process of measured variables being observed or missing. The pooled point and $SE$ estimates can be used to calculate pooled Wald-type statistics [49,50]; various methods have also been proposed for pooled score [51] and likelihood-ratio test (LRT) [52,53] statistics, which are asymptotically equivalent [54].

Mislevy et al. [2] proposed capitalizing on the multiple-imputation framework to incorporate IRT person parameters as variables in standard regression-based analyses (i.e., two-stage estimation). Latent person parameters are sampled from the posterior (along with estimated item parameters), and each posterior sample is treated as an imputation of the unobserved person parameters. A regression model can be fitted to each imputation, and results are pooled across plausible-value samples so that inferences account for uncertainty more appropriately than analyzing a single point estimate of each IRT person parameter. This approach is also available to improve inferences based on factor-score regression [55,56].

Lüdtke et al. [7] demonstrated how plausible values of SRM effects can be sampled using MCMC and included in a person-level regression model. MCMC estimation typically involves drawing several hundreds or thousands of samples from the posterior to draw inferences about parameters, but only a few such posterior samples (spaced out or "thinned" to avoid autocorrelation and minimize computational burden) would be used as multiple

imputations. By embedding case-level SRM effects in a more complex multivariate system of variables, the case-level model of an SR-SEM could feasibly be estimated using the plausible-values approach ([7] p. 119). The approach we describe in Section 2 utilizes the full posterior distribution of estimated summary statistics to improve stability relative to a small posterior sample of person-level parameter estimates. The plausible-values approach follows a similar two-stage estimation procedure (first estimating latent components, then using them as data in a subsequent model), but it analyzes case-level data in Stage 2 estimation, whereas our limited-information approach analyzes summary statistics in Stage 2 estimation.

*1.2. Structural Equation Modeling*

We introduce SEM via the so-called "all-y" LISREL parameterization employed by `lavaan` [57] and M*plus* [58], which is composed of measurement and structural components. In the measurement model, $V$ observed "indicator" variables in the vector $y$ are a linear function of $F \leq V$ latent "factor" variables in the vector $\eta$, and the structural component allows latent variables to be regressed on each other:

$$y = \nu + \Lambda\eta + \varepsilon, \tag{16}$$
$$\eta = \alpha + B\eta + \zeta. \tag{17}$$

The measurement model in Equation (16) includes a vector of $V$ intercepts $\nu$; a $V \times F$ matrix of factor loadings (linear regression slopes: $\Lambda$) relating indicators to latent variables; and a vector of $V$ indicator residuals $\varepsilon$. The structural model in Equation (17) includes a vector of $F$ latent-variable intercepts $\alpha$; an $F \times F$ matrix of linear regression slopes relating latent variables to each other ($B$, with diagonal constrained to zero); and a vector of $F$ residuals $\zeta$. Indicator residuals are assumed to be uncorrelated with latent variables and latent residuals, but each may covary among themselves and are assumed multivariate normally distributed:

$$\varepsilon \sim \mathcal{MVN}(0, \Theta) \text{ and } \zeta \sim \mathcal{MVN}(0, \Psi). \tag{18}$$

Means and (co)variances of observed indicators are assumed to be structured as a function of SEM parameters:

$$
\begin{aligned}
\mathrm{E}(y) &= \mu(\vartheta) = \nu + \Lambda(I - B)^{-1}\alpha, \\
\mathrm{Var}(y) &= \Sigma(\vartheta) = \Lambda(I - B)^{-1}\Psi(I - B)^{-1'}\Lambda' + \Theta,
\end{aligned}
\tag{19}
$$

with identity matrix $I$ whose dimensions match $B$. SEM parameters to be estimated are collected in a single vector $\vartheta$, and sample estimates $\mu(\hat{\vartheta}) = \hat{\mu}$ and $\Sigma(\hat{\vartheta}) = \hat{\Sigma}$ of the model-implied moments in Equation (19) are obtained by plugging in sample estimates of the corresponding parameters. Various least-squares and ML estimators of $\hat{\vartheta}$ minimize the overall discrepancy between observed summary statistics ($\bar{y}$ and $S$) and corresponding model-implied moments ($\hat{\mu}$ and $\hat{\Sigma}$) [59]. Thus, under certain conditions (e.g., multivariate normally distributed, no missing observations), SEM parameters can be estimated using summary statistics as input rather than raw casewise data.

This section concludes by describing SEM for round-robin data (Section 1.2.3). However, first, there are two subsections that provide some context for how two-stage estimation algorithms have been used to apply SEM to other complex data conditions. These algorithms inspired the current proposal, described in Section 2.

1.2.1. Efficient SEM with Multiple Imputations

Fitting a specified SEM to multiple imputed data sets can become quite computationally intensive for large, complex models. The ability to fit SEMs to summary statistics motivated Cai and colleagues [10,11] to develop a less computationally intensive alternative. One can obtain pooled estimates of $\bar{y}$ and $S$ from multiple imputations by fitting a

"saturated" model (unrestricted covariance matrix and mean vector), then averaging the saturated-model estimates $\hat{\mu}_m$ and $\hat{\Sigma}_m$ across imputations $m = 1, \ldots, M$ (i.e., Rubin's rules). The pooled estimates can then be used as input data to fit any hypothesized SEM(s) only once, rather than $M$ times. Although this would yield unbiased parameter estimates (given satisfied distributional assumptions), there are two problems with this approach, for which Cai and colleagues [10,11] proposed solutions.

The first problem with this approach is that the *SE*s of $\hat{\vartheta}$ would be underestimated when relying on standard approaches that treat $\hat{\mu}$ and $\hat{\Sigma}$ as observed summary statistics ($\bar{y}$ and $S$). The proposed solution [10] is to also apply Rubin's rules to the (squared) *SE*s of the estimated moments—or rather, to the entire asymptotic covariance matrix (ACOV) of estimated parameters. The within-imputation component of the pooled ACOV is obtained by averaging ACOV$_m$ across $M$ imputations, and the between-imputation component of the pooled ACOV is obtained by calculating (co)variances of point estimates across imputations. The pooled ACOV can then be multiplied by the sample size $N$ and used as the "gamma" matrix ($\Gamma = N \times$ ACOV) in standard formulas for robust *SE*s (see Lee and Cai, 2012 [10], p. 686, or Savalei, 2014 [60], Equation (14)). Computational formulas for estimating $\Gamma$ with (in)complete (non-)normal data Savalei and Rosseel (2022 [61], e.g., Equation (24) on p. 167), along with computational formulas for ACOV ([61] p. 168, Equation (34) and Table 2).

The second problem with this approach (related to the first) is that a standard LRT statistic of data–model fit would be overestimated. The proposed solution [10] is to calculate a residual-based test statistic ([59] Equation (2.20a))

$$T_{\text{res}} = N \times e' \Gamma^{-1} e, \tag{20}$$

where $e = [\hat{\mu}, vech(\hat{\Sigma})]' - [\bar{y}, vech(S)]'$ is the vector of $V^*$ mean and covariance residuals, $V^* = \frac{V(V+3)}{2}$ is the number of sample means and (co)variances, and $vech(.)$ is a half-vectorizing function that yields nonredundant elements of a covariance matrix. Under the $H_0$, $T_{\text{res}}$ is asymptotically $\chi^2(df)$ distributed with $df$ equal to $V^*$ minus the number of estimated SEM parameters in $\hat{\vartheta}$. Simulation studies [10,11] showed $T_{\text{res}}$ maintained Type I error rates close to nominal levels for this two-stage approach. Note that $T_{\text{res}}$ utilizes the same $\Gamma$ matrix used to correct the *SE* (see Lee and Cai, 2012 [10], p. 686).

Lee and Cai's [10] two-stage estimator was developed using MLE to obtain Stage 1 estimates. After introducing the SR-SEM in Section 1.2.3, we then describe in Section 2 how their two-stage estimator can be adapted to work with MCMC estimation in Stage 1. However, first, we briefly present multilevel SEM (ML-SEM) to introduce the idea of using SEM to model different levels of analysis.

### 1.2.2. Multilevel Structural Equation Model

When primary sampling units (e.g., persons $i \in 1, \ldots, N_g$) are clustered within groups $g \in 1, \ldots, G$ (e.g., students nested in schools), observed data $y_{ig}$ will not be independent observations because clustering introduces dependence among cases within the same group. This violates the independence assumption of standard least-squares and ML estimators of single-level data. The dependence in $y_{ig}$ can be accounted for by disentangling the between-cluster components (i.e., cluster means $\bar{y}_g$) and the within-cluster components (i.e., cluster-mean centered $y_{ig} - \bar{y}_g$). Optionally, the grand mean ($\bar{y}$) could be further partitioned from cluster means:

$$\begin{aligned} y_{ig} &= (y_{ig} - \bar{y}_g) + \bar{y}_g \\ &= (y_{ig} - \bar{y}_g) + (\bar{y}_g - \bar{y}) + \bar{y}. \end{aligned} \tag{21}$$

In the multivariate case, this generalizes to calculating cluster-specific mean vectors (between-cluster components) to cluster-mean center the vector $y_{ig}$. The within-cluster components of variables $v = 1, \ldots, V$ are distributed around a mean vector $\mathbf{0}$ with covariance matrix $S_W$, and the between-cluster components are distributed around grand mean

vector $\mu$ with covariance matrix $S_B$. The total covariance matrix $S_T$ of the composite vector $y_{ig}$ is the sum of these two orthogonal components:

$$S_T = S_W + S_B. \tag{22}$$

Structural equation models for clustered data were first introduced [62] by conducting the multivariate decomposition in Equation (21), calculating level-specific covariance matrices in Equation (22), and fitting a level-specific SEM to the covariance matrix of its corresponding level of analysis. Both levels can be modeled simultaneously by treating the two level-specific covariance matrices as though they came from independent groups [63], but tedious constraints must be specified to ensure the (log-)likelihood is correct. This is due to the fact that $S_W$ and $S_B$ are not consistent estimators of their corresponding population counterparts $\Sigma_W$ and $\Sigma_B$ (although $S_W$ would be a consistent estimator when all cluster sizes $N_g$ are equal; see Muthén, 1994 [63], p. 384 for details).

Analysis of casewise observations allows for less problematic full-information estimators [64,65], although wide-format analysis of clusterwise observations [66–68] can be applied when feasible (e.g., many small clusters, few variables). However, there are advantages to fitting SEMs separately to each level of analysis, such as reduced computational complexity and separately evaluating data–model correspondence at each level (see [69] about the partially saturated approach to evaluating ML-SEM fit). Yuan and Bentler [6] proposed consistent estimates of decomposed $\Sigma_W$ and $\Sigma_B$, as well as their associated ACOVs, which are used to obtain accurate $SE$s and a residual-based statistic based on Browne [59]. This two-stage approach to ML-SEM is conceptually similar to Cai and colleagues' [10,11] two-stage approach for SEM with missing data and thus is also analogous to our proposal for round-robin data.

### 1.2.3. Social Relations Structural Equation Model

Round-robin data follow a more complex nesting structure than the two-level example discussed in the previous section. Dyadic observations $\mathbf{y}_{ij}$ are cross-classified, nested under the same cases in two ways: all dyads containing case $i$ are nested under ego (actor or perceiver) $i$, and all dyads containing case $j$ are nested under alter (partner or target) $j$. The multivariate SRM in Section 1.1.1 thus decomposes a covariance matrix of dyadic observations $\Sigma_y$ into case- and dyad-level components (and a group-level component, if cases $\mathbf{y}_{g\{ij\}}$ are additionally nested in multiple round-robin groups):

$$\Sigma_y = \Sigma_G \otimes \begin{bmatrix} 1 & 1 \\ 1 & 1 \end{bmatrix} + \Sigma_{EA} + \Sigma_{AE} + \Sigma_R, \tag{23}$$

where a Kronecker product $\otimes$ is used to duplicate the elements of $\Sigma_G$ from Equation (6) so the dimensions match $\Sigma_{EA}$ from Equation (7) and $\Sigma_R$ from Equation (8). The case-level matrix $\Sigma_{AE}$ is a rearrangement of $\Sigma_{EA}$, such that components are ordered [Alter, Ego, ..., Alter, Ego] rather than [Ego, Alter, ..., Ego, Alter]:

$$\Sigma_{AE} = \begin{bmatrix} \sigma^2_{A_1} & & & \\ \sigma_{E_1,A_1} & \sigma^2_{E_1} & & \\ \sigma_{A_2,A_1} & \sigma_{A_2,E_1} & \sigma^2_{A_2} & \\ \sigma_{E_2,A_1} & \sigma_{E_2,E_1} & \sigma_{E_2,A_2} & \sigma^2_{E_2} \end{bmatrix} \tag{24}$$

Thus, similar to specifying separate SEMs for each level of multilevel data [6], a distinct SEM can be specified for each (group-, case-, or dyad-level) covariance matrix.

Only the group level contains mean-structure parameters (intercepts):

$$\mu_g = \nu + \Lambda^{(\mathbf{G})}\eta_g^{(\mathbf{G})} + \varepsilon_g^{(\mathbf{G})}, \tag{25}$$

$$\eta_g^{(\mathbf{G})} = \alpha + B^{(\mathbf{G})}\eta_g^{(\mathbf{G})} + \zeta_g^{(\mathbf{G})}, \tag{26}$$

where $\mu_g$ contains the group components (Equation (6)); intercepts $\nu$ and $\alpha$ are defined as they were for Equations (16) and (17); and the group-level latent variables in $\eta_g^{(\mathbf{G})}$, group-level residuals in $\varepsilon_g^{(\mathbf{G})}$ and $\zeta_g^{(\mathbf{G})}$, and group-level slopes in $\Lambda^{(\mathbf{G})}$ and $B^{(\mathbf{G})}$ are defined analogously for group components as for single-level quantities in Equations (16) and (17). Analogous distributional assumptions are also made about group-level residuals:

$$\varepsilon_g^{(\mathbf{G})} \sim \mathcal{MVN}(0, \Theta^{(\mathbf{G})}) \text{ and } \zeta_g^{(\mathbf{G})} \sim \mathcal{MVN}(0, \Psi^{(\mathbf{G})}). \tag{27}$$

Group-level model parameters (collected in vector $\vartheta^{(\mathbf{G})}$) imply a structure for the group-level means and (co)variances in Equation (6):

$$\begin{aligned}
\mu(\vartheta^{(\mathbf{G})}) &= \nu + \Lambda^{(\mathbf{G})}(I - B^{(\mathbf{G})})^{-1}\alpha, \\
\Sigma(\vartheta^{(\mathbf{G})}) &= \Lambda^{(\mathbf{G})}(I - B^{(\mathbf{G})})^{-1}\Psi^{(\mathbf{G})}(I - B^{(\mathbf{G})})^{-1'}\Lambda^{(G)'} + \Theta^{(\mathbf{G})},
\end{aligned} \tag{28}$$

where $\mu(\vartheta^{(\mathbf{G})})$ and $\Sigma(\vartheta^{(\mathbf{G})})$ are the model-implied analogs to saturated-model parameters $\mu_G$ and $\Sigma_G$, respectively, in Equation (6).

Analogously, group $g$'s case-level components $E_{gi}$ and $A_{gi}$ of each variable are stored in a vector $\mathbf{u}_{gi}$, arranged as in Equation (7). These components have their own measurement and structural models:

$$\mathbf{u}_{gi} = \Lambda^{(\mathbf{u})}\eta_{gi}^{(\mathbf{u})} + \varepsilon_{gi}^{(\mathbf{u})}, \tag{29}$$

$$\eta_{gi}^{(\mathbf{u})} = B^{(\mathbf{u})}\eta_{gi}^{(\mathbf{u})} + \zeta_{gi}^{(\mathbf{u})}, \tag{30}$$

where case-level common factors in $\eta_{gi}^{(\mathbf{u})}$ are also expected to have ego and alter components [8,9], as are case-level residuals in $\varepsilon_{gi}^{(\mathbf{u})}$ and $\zeta_{gi}^{(\mathbf{u})}$, which are distributed with case-level covariance matrices:

$$\varepsilon_{gi}^{(\mathbf{u})} \sim \mathcal{MVN}(0, \Theta^{(\mathbf{u})}) \text{ and } \zeta_{gi}^{(\mathbf{u})} \sim \mathcal{MVN}(0, \Psi^{(\mathbf{u})}). \tag{31}$$

Case-level model parameters imply a structure for the case-level covariance matrix (the model-implied analog to saturated-model parameter $\Sigma_{EA}$ in Equation (7)):

$$\Sigma(\vartheta^{(\mathbf{u})}) = \Lambda^{(\mathbf{u})}(I - B^{(\mathbf{u})})^{-1}\Psi^{(\mathbf{u})}(I - B^{(\mathbf{u})})^{-1'}\Lambda^{(\mathbf{u})'} + \Theta^{(\mathbf{u})}. \tag{32}$$

Finally, group $g$'s dyad-level (or "relationship"-level) components $R_{gij}$ and $R_{gji}$ for each variable are stored in a vector $\mathbf{r}_{gd}$, arranged as in Equation (8). These components have their own measurement and structural models:

$$\mathbf{r}_{gd} = \Lambda^{(\mathbf{r})}\eta_{gd}^{(\mathbf{r})} + \varepsilon_{gd}^{(\mathbf{r})}, \tag{33}$$

$$\eta_{gd}^{(\mathbf{r})} = B^{(\mathbf{r})}\eta_{gd}^{(\mathbf{r})} + \zeta_{gd}^{(\mathbf{r})}, \tag{34}$$

where dyad-level common factors in $\eta_{gd}^{(\mathbf{r})}$ follow the same pattern as dyad-level residuals in $\varepsilon_{gd}^{(\mathbf{r})}$ and $\zeta_{gd}^{(\mathbf{r})}$, which are distributed with dyad-level covariance matrices:

$$\varepsilon_{gd}^{(\mathbf{r})} \sim \mathcal{MVN}(0, \Theta^{(\mathbf{r})}) \text{ and } \zeta_{gd}^{(\mathbf{r})} \sim \mathcal{MVN}(0, \Psi^{(\mathbf{r})}). \tag{35}$$

The dyad-level covariance matrices are subject to the same equality constraints depicted in Equation (8); likewise, dyad-level slopes in $\Lambda^{(\mathbf{r})}$ and $B^{(\mathbf{r})}$ have equality constraints consistent with indistinguishable dyads (see example in Section 2). These constraints on

dyad-level model parameters therefore yield a model-implied structure for the dyad-level covariance matrix that is analogous to the saturated-model parameter $\Sigma_R$ in Equation (8):

$$\Sigma(\vartheta^{(\mathbf{r})}) = \Lambda^{(\mathbf{r})}(I - B^{(\mathbf{r})})^{-1}\Psi^{(\mathbf{r})}(I - B^{(\mathbf{r})})^{-1'}\Lambda^{(\mathbf{r})'} + \Theta^{(\mathbf{r})}. \tag{36}$$

In the next section, we describe our proposed two-stage (limited-information) estimator for the SR-SEM described in this section. However, first, we point out how our SR-SEM formulation differs from the one first proposed by Nestler et al. [8], who proposed a single-stage FIML estimator to obtain point and *SE* estimates. Our SR-SEM is formulated to meet the needs of our data example, so it differs from their SR-SEM in at least two ways.

First, Nestler et al. [8] included effects of fixed exogenous covariates *X* on common factors at case and dyad levels, which we exclude here for brevity and lack of relevance to the data example. In principle, an SR-SEM could include exogenous-variable effects on both indicators and common factors [70], and we speculate in Section 4 how our two-stage estimator could be adapted for this extension.

Second, their model excluded a group level, which we include to demonstrate a possible interest in modeling group-level (co)variances. When there is only one round-robin group, the group-level covariance matrix $\Sigma_G = \mathbf{0}$, and the mean structure could be modeled at the case level (e.g., as means of ego or of alter effects, as Nestler et al. [8,9] chose to represent means).

## 2. Materials and Methods

Our two-stage algorithm for estimating SR-SEM parameters was inspired by previous two-stage SEM estimation algorithms, such as Cai and colleagues' [10,11] method described in Section 1.2.1 and the ML-SEM method of Yuan and Bentler [6] described in Section 1.2.2. Both of these earlier two-stage estimators used a limited-information frequentist estimator in Stage 2 (as we do), but they also used frequentist (e.g., ML or least-squares) estimators in Stage 1. Our algorithm employs MCMC estimation for Stage 1, similar to the plausible-values approach [7] or multiple imputation in general (see Section 1.1.3). Although MCMC is typically used to draw inferences from a Bayesian perspective [15,31], we use MCMC simply to rely on its flexibility in capturing joint uncertainty of SRM-component (co)variances among many variables. In principle, MLE could be used for Stage 1 estimation (e.g., by fitting a saturated SR-SEM via the `srm` package [71]), a possibility we discuss in Section 4.

We show how our algorithm can be implemented using existing open-source software packages in the R [72] environment, and we demonstrate the method using open-access round-robin data [12,13]. We provide some relevant syntax in our Appendices, and all syntax and data files are provided in this paper's companion OSF project (https://osf.io/2qs5w/).

### 2.1. Example Data

We begin by briefly describing the data [12], so that we can describe our algorithm while providing the context of a concrete accompanying example. Salazar Kämpf et al. [13] investigated the mediating role of social mimicry in the development of liking before and after getting acquainted. German university students ($N = 139$) were assigned to 26 small ($N_g \in 4$–6) same-sex groups of strangers. Preinteraction liking was measured on a 1–6 Likert scale (higher scores indicated more liking) with 2 indicators ("I like this person" and "I would like to get to know this person"). Within each group, students had a 5 min interaction with each other group member, which was video-recorded so that three independent observers could rate (on a 1–6 Likert scale) how much each participant mimicked their conversation partner during the interaction. Postinteraction liking was measured by the same two indicators as preinteraction liking, as well as by a third indicator ("I would like to become friends with this person").

We calculated composite scores to represent each construct: *liking* was the average score across two (preinteraction) or three (postinteraction) indicators, with higher scores indicating more liking. Likewise, the observed variable *mimicry* was the average score

across the three raters, with higher scores indicating more social mimicry. Rather than calculating composite scores for each construct, Salazar Kämpf et al. [13] used multivariate SRM to disentangle relationship effects from dyad-level measurement error, so their path-model results are not directly comparable to the path-model results we provide using FIML or two-stage MLE. Although the SR-SEM opens the possibility of modeling these three variables as common factors (each with 2–3 indicators), we opted to keep the example application simpler by modeling composite scores.

Salazar Kämpf et al. [13] used univariate regression to estimate parameters of the dyad-level model depicted in Figure 1 (comparable to their Figure 1 on p. 135). Note that the paths among the lower three components are constrained to equality with the paths among the upper three components, as implied by indistinguishable dyads. Residuals for social mimicry are omitted from Figure 1 due to space constraints.

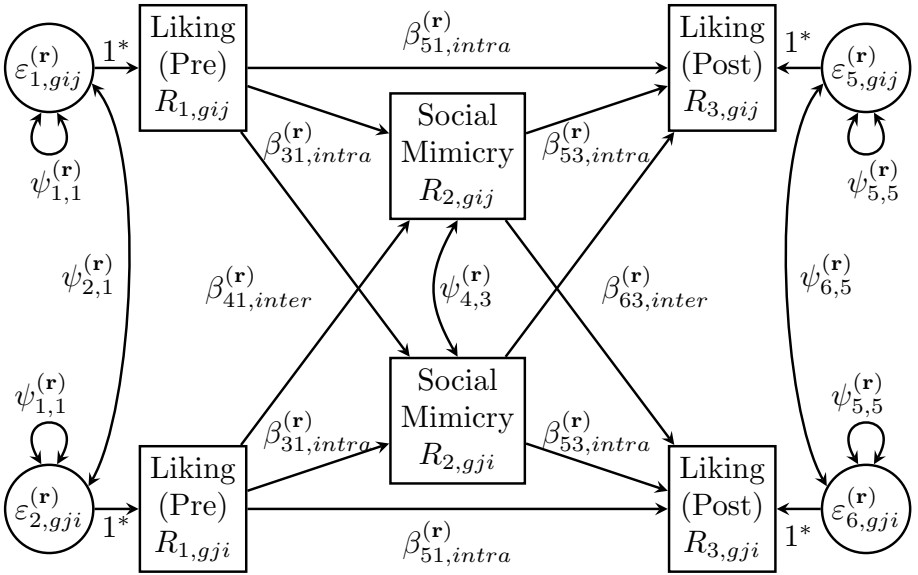

**Figure 1.** Path diagram depicting hypothesized [13] dyad-level causal process. Equality constraints consistent with indistinguishable dyads are reflected by using the same label for a pair of parameters. Residuals for social mimicry (and their variance: $\psi_{3,3}^{(\mathbf{r})} = \psi_{4,4}^{(\mathbf{r})}$) are omitted to save space.

We also fit an analogous path model at the case level, depicted in Figure 2. Salazar Kämpf et al. [13] also estimated most of these paths in separate univariate regression models (presented in their online supplements [12]), but their report focused primarily on the dyad-level results. Note that we do not specify equality constraints for this model, as we did for the dyad level in Figure 1, because ego and alter effects are distinguishable. For the liking variables, these are perceiver and target effects, respectively. For social mimicry (measured by behavioral observation), these are actor and partner effects, respectively.

Salazar Kämpf et al. [13] did not have hypotheses about the group level of analysis, and the SR-SEM of Nestler et al. [8] does not include a group-level component. Although we can expect little group-level variance in their design due to the random assignment of subjects to groups [13], we think it would be reasonable to hypothesize that groups displaying more social mimicry would also elicit more postinteraction liking, on average, across all interactions. For the sake of completeness, we additionally fit a group-level path model, depicted in Figure 3. Note that there is only one component per variable (the group average).

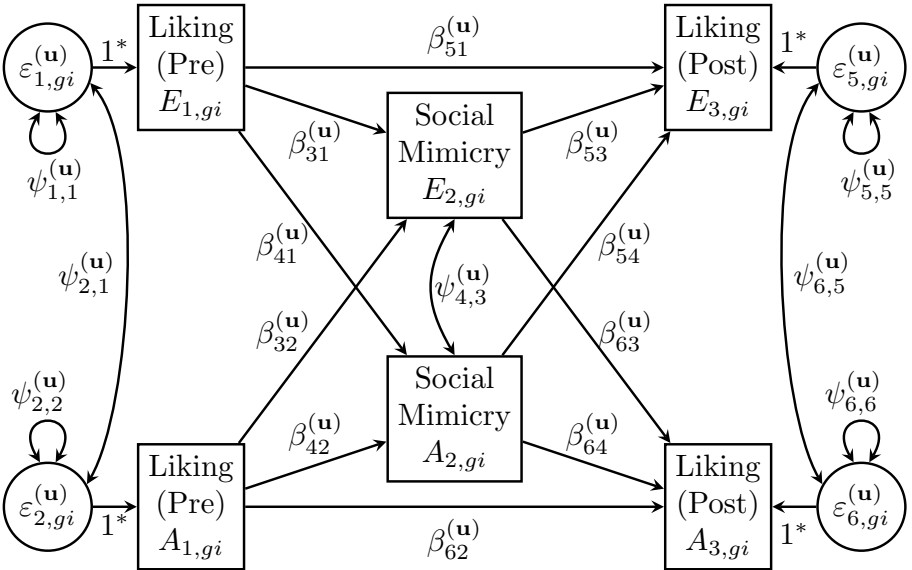

**Figure 2.** Path diagram depicting case-level causal process. Residual variances for social mimicry ($\psi_{3,3}^{(\mathbf{u})}$ and $\psi_{4,4}^{(\mathbf{u})}$) are omitted to save space.

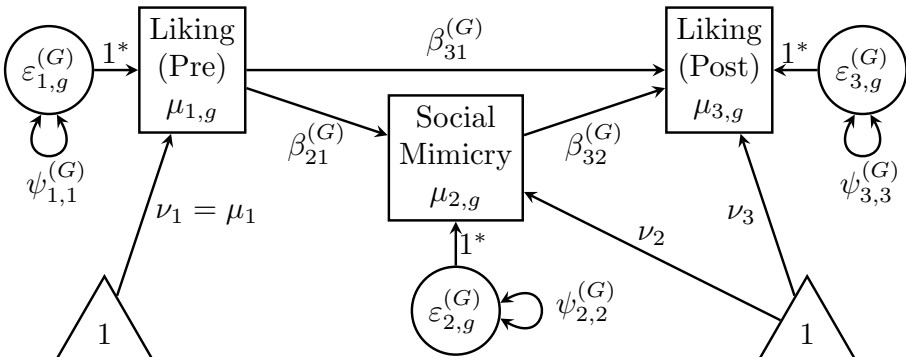

**Figure 3.** Path diagram depicting group-level causal process.

### 2.2. Two-Stage Estimation of SR-SEM Parameters

Our two-stage method begins by estimating multivariate SRM parameters to obtain point estimates of covariance matrices at each level of interest (group, case, dyad), as well as estimates of their sampling (co)variances (i.e., ACOV). Stage 1 point estimates are then treated as input data to estimate SR-SEM parameters in Stage 2.

#### 2.2.1. Stage 1: Estimate Multivariate SRM Parameters

Whereas Nestler [30] proposed a Fisher-scoring algorithm to obtain ML estimates of multivariate SRM parameters (see Equations (6)–(8)), Stage 1 of our two-stage method utilizes MCMC estimation [15,31,33] by employing a modified Hamiltonian Monte Carlo algorithm known as the No-U-Turn Sampler (NUTS) [73], available in the general Bayesian modeling R package `rstan` [74,75]. Unlike other MCMC methods such as Gibbs sampling, the NUTS simultaneously samples the entire vector of all unknown parameters from a multidimensional parameter space. A practical advantage over Gibbs sampling is that priors do not need to be conjugate, so researchers can specify prior distributions that are intuitive to interpret.

The unknown parameters in Stage 1 included the round-robin variable means, the level-specific random effects, and the *SD*s of—and correlations among—the random effects. We used weakly informative priors to estimate the SRM distributional parameters (i.e., the mean vector and level-specific covariance matrices). Given the scale of the observed data

(1–6 Likert scale, median of possible values = 3.5), we specified priors for the round-robin variable means as normal with unit variance, centered at $\mu = 3.5$:

$$\mu_G \sim \mathcal{N}(\mu = 3.5, \sigma = 1). \tag{37}$$

This prior reflects the belief that the mean is most likely to be in the range of 2.5–4.5 (68% of probability mass) and is relatively unlikely (though not impossible) to be near the endpoints of the scale. Viewing histograms of the round-robin variables easily confirms this is not unreasonable for these data.

As advised by Gelman [76] and recommended by the Stan developers (who maintain a web page with recommendations and further reading about specifying priors: https://github.com/stan-dev/stan/wiki/Prior-Choice-Recommendations#prior-for-scale-parameters-in-hierarchical-models, accessed on 27 January 2024), the *SD*s of level-specific random effects were specified with Student's $t_{df=4}$ distributions, truncated below at zero (so *SD*s could not be negative). The kurtosis ($\frac{6}{df-4}$) is undefined due to division by 0 (thus approaches infinity in the limit), making it less restrictive than a truncated normal distribution about values being much larger than the mean. Simulation research has shown that this prior works well in variance-decomposition models [77], and it is the default prior for scale parameters in the R package brms [78]. Stan provides location ($\mu$) and scale ($\sigma$) parameters, to center the *t* distribution at a different mean or to have greater variance. We selected scaling parameters for each variance component to reflect our prior belief (based on most empirical SRM research) that the majority of variance would be relationship-specific (i.e., at the dyad level). Given that the total variance of each round-robin variable was slightly greater than 1 (min = 1.05, max = 1.144), a *t* distribution centered at 0.5 places a mode at nearly 50% of the total variance, but a scaling parameter of 0.5 still allows for a high prior probability that the relationship variance is as little as 0 or as large as 1 (which would be nearly 100% of the observed variance):

$$\sigma_R \sim t(df = 4, \mu = 0.5, \sigma = 0.5). \tag{38}$$

Given that the groups were formed by random assignment, we expected very little group-level variance, and so the prior mean of $\sigma_G$ was specified as quite small:

$$\sigma_G \sim t(df = 4, \mu = 0.05, \sigma = 0.5), \tag{39}$$

whereas we expected the person-level effects to have larger variance components (thus, higher prior means):

$$\sigma_E \text{ and } \sigma_A \sim t(df = 4, \mu = 0.25, \sigma = 0.5). \tag{40}$$

Priors for all *SD*s were specified with scaling factors of $\sigma = 0.5$, making them only weakly informative because no variance component was precluded from being as small as 0 or as large as 1.

Priors for the group- and case-level correlation matrices ($\Sigma^*$, i.e., standardized covariance matrices $\Sigma$) were specified to follow an LKJ distribution [79]:

$$\Sigma_G^* \text{ and } \Sigma_{EA}^* \sim LKJ(\eta = 2) \tag{41}$$

A shape parameter $\eta = 1$ would imply a uniform distribution (i.e., all correlation matrices are equally likely, with values spanning $\pm 1$), whereas higher values of $\eta > 1$ correspond to distributions with a mode at the identity matrix (i.e., correlations of zero) and correlations distributed symmetrically around zero. Thus, a value of $\eta = 2$ is close to a uniform distribution but with a slightly higher probability of correlations being smaller than larger in absolute value. This expectation conforms to most published SRM results, where large correlations ($r > 0.5$) are much rarer than small-to-moderate correlations. However,

the "low" mode at the identity matrix is only weakly informative, so the posterior is overwhelmingly influenced by the data.

The dyad-level correlation matrix has several equality constraints reflecting indistinguishable dyads, whereas an LKJ prior's only restriction is that the correlation matrix is positive definite. Given $V$ round-robin variables, the number of unique covariances $\Sigma_G$ (=correlations in $\Sigma_G^*$) would be $\frac{V(V-1)}{2}$ and the number of unique covariances in $\Sigma_{EA}$ would be $\frac{2V(2V-1)}{2}$. However, the number of unique covariances in $\Sigma_R$ would be $V + V(V-1)$, consisting of $V$ dyadic reciprocities (one per variable), $\frac{V(V-1)}{2}$ *intra*personal correlations between variables, and $\frac{V(V-1)}{2}$ *inter*personal correlations between variables (e.g., Equation (8)). An LKJ prior would not allow for these equality constraints, so a prior must be specified to sample each correlation separately, which are then scaled to covariances and placed into their appropriate positions in $\Sigma_R^*$. To maintain the requirement that $\Sigma_R^*$ is positive definite, Lüdtke et al. [7] proposed placing constraints on the determinants of its principle minors, as described by [80]. Although this was shown to be feasible for bivariate models when using Gibbs sampling via WinBUGS [7,80], specifying such constraints for multivariate SRMs with arbitrary $V > 2$ becomes infeasibly tedious. Luckily, the `blavaan` package [81] has already demonstrated that the adaptive nature of Stan's NUTS algorithm tends to "learn" quickly during the warm-up samples to avoid non-positive-definite (NPD) corners of the parameter space that lead to rejecting the sample.

Inspired by the priors used for correlations in the `blavaan` package [81], rescaled Beta distributions were specified as priors for all correlations at the dyad level:

$$\rho_R, \rho_{\text{intra}}, \text{ and } \rho_{\text{inter}} \sim \text{Beta}_{-1,1}(\alpha = 1.5, \beta = 1.5). \tag{42}$$

A $\text{Beta}_{-1,1}(.)$ distribution provides support on the $\{-1,1\}$ scale (rather than the usual $\{0,1\}$ scale) by multiplying a sampled value by 2 then subtracting 1. A Beta prior with $\alpha = \beta = 1$ corresponds to a perfectly uniform distribution, where larger shape parameters imply sharper peaks, remaining symmetric as long as $\alpha = \beta$ (implying the highest prior density at a correlation of 0). Our chosen shape parameters in Equation (42) were 1.5, yielding very little density for extremely large absolute values of correlations. Specifically, the central 60% of probability density for a $\text{Beta}_{-1,1}(1.5, 1.5)$ distribution captures correlations of $\pm 0.5$, and the central 90% captures correlations of $\pm 0.8$. One can visualize this distribution with the R syntax: `curve(dbeta((x+1)/2, 1.5, 1.5), from = -1, to = 1)`. Thus, similar to the LKJ priors, the parameters of these Beta priors were specified such that there was a "low" mode at correlations of zero but were similar enough to a uniform distribution spanning $\pm 1$ that posterior distributions were overwhelmingly influenced by the data.

Although it is possible to sample combinations of correlations that yield NPD correlation matrices, such samples are less likely given the slight prior restrictions on very large correlation values. Nonetheless, NPD matrices were frequent enough during the adaptation phase that the algorithm would fail before sampling. To avoid this, we chose to randomly sample starting values from a smaller range—drawn from a $U(-0.5, +0.5)$ distribution—rather than from Stan's default $U(-2, +2)$ distribution, using the argument `init_r = 0.5` (see Appendix B).

Hyperpriors for random effects parameters at the group and case levels were effectively the covariance matrices implied by the estimated correlations and *SD*s (see Equations (6) and (7), respectively). However, as the Stan syntax in Appendix A reveals, we achieved greater stability and computational efficiency in two ways. First, we used Stan's Cholesky parameterization of the multinormal sampling function, which uses a Cholesky decomposition of the correlation matrix that we calculated in our Stan syntax. Second, we sampled random effects from multivariate standard-normal distributions (i.e.,

with unit variance). The sampled random effects were then multiplied by their respective *SD*s when calculating expected values $\widehat{\mathbf{y}}_{g\{ij\}}$ for each dyadic observation:

$$\widehat{\mathbf{y}}_{g\{ij\}} = \begin{bmatrix} \hat{y}_{1,gij} \\ \hat{y}_{1,gji} \\ \vdots \\ \hat{y}_{k,gij} \\ \hat{y}_{k,gji} \end{bmatrix} = \begin{bmatrix} \mu_{1,g} \\ \mu_{1,g} \\ \vdots \\ \mu_{k,g} \\ \mu_{k,g} \end{bmatrix} + \begin{bmatrix} E_{1,gi} \\ A_{1,gi} \\ \vdots \\ E_{k,gi} \\ A_{k,gi} \end{bmatrix} + \begin{bmatrix} A_{1,gj} \\ E_{1,gj} \\ \vdots \\ A_{k,gj} \\ E_{k,gj} \end{bmatrix}. \tag{43}$$

Finally, each dyadic observation's likelihood was specified as a multivariate normal distribution with mean vector equal to the expected values in Equation (43):

$$\mathbf{y}_{g\{ij\}} \sim \mathcal{MVN}(\mu = \widehat{\mathbf{y}}_{g\{ij\}}, \Sigma_R) \tag{44}$$

where $\Sigma_R$ is the dyad-level covariance matrix. As with the random effect hyperpriors, our Stan syntax in Appendix A uses the Cholesky parameterization for computational efficiency. Note that when the vector of observations $\mathbf{y}_{g\{ij\}}$ is incomplete, the likelihood in Equation (44) is also the prior for missing data that are sampled from the posterior (i.e., data augmentation; [31,33]).

Four chains of 5000 iterations—2500 burn-in and 2500 posterior samples—each were used to estimate the joint posterior distribution of model parameters, yielding 10,000 posterior samples for inference. Initial values for each Markov chain in Stage 1 were randomly sampled from a uniform distribution. Convergence was diagnosed by inspecting traceplots to verify adequate mixing. Effective posterior sample sizes were sufficiently large for all parameters (range: 282–10,100), and all parameters had a sufficiently small potential scale-reduction factor (PSRF [82] or $\hat{R} \leq 1.01$). The multivariate PSRF [83] was 1.03, which we deemed sufficiently small for the purposes of this demonstration. See Appendix B for R syntax to fit the Stan model in Appendix A.

2.2.2. Prepare Stage 1 Results for Stage 2 Input

Following MCMC estimation in `rstan`, we calculated the covariance matrix implied by the estimated correlations and *SD*s in each posterior sample (see Appendix B.1). Point estimates were computed as the average of the estimated (co)variances across posterior samples—i.e., expected a posteriori (EAP) estimates. This is analogous to Rubin's rules for pooled estimates across multiple imputations of missing data. The missing values in our example are latent variables (i.e., random effects, which are missing for all sampling units), and every posterior sample imputes (or "augments") those missing values by sampling them from the posterior. However, our interest is not in those plausible-value estimates but in the summary statistics used as their hyperparameters. Thus, the Stage 1 output of interest includes three level-specific covariance matrices between the SRM components (and group-level means). Interpreting these multivariate SRM parameters might also be of substantive interest, in which case uncertainty about estimates can be quantified by computing credible intervals from the empirical posterior distribution.

Next, we prepare Stage 1 results for ML estimation of SR-SEM parameters in Stage 2 by adapting methods described in Section 1.2.1 for efficient SEM estimation with multiple imputations. We must prepare point estimates (EAPs, as described above) of level-specific summary statistics to use as input data. To obtain *SE*s and test statistics that account for the uncertainty of the estimated summary statistics, we must calculate $\Gamma = N \times \text{ACOV}$, as we describe next.

When using multiple imputations of missing data, Section 1.1.3 explained that two sources of uncertainty (complete- and missing-data sampling variance) must be pooled. This is because imputed data sets are analyzed using complete-data statistical methods after the incomplete data are imputed using a separate statistical model. The missing-data component (*B*) then needs to be added to the complete-data component (*W*) of

overall sampling variance. Using MCMC makes this unnecessary because the missing data (random effects) are augmented during estimation of the parameters of interest, so the imputation and analysis models are concurrent.

Therefore, an estimate of the ACOV matrix is obtained simply by computing the posterior sampling (co)variances between the SRM mean and (co)variance parameters estimated in Stage 1. The calculation is conducted by arranging a data matrix with SRM parameters of interest in columns, and each row contains a sample from the posterior distribution. We then use scalar multiplication to obtain $\hat{\Gamma}$ by multiplying the level-specific ACOV by that level's number of observations. Appendix B.1 shows how the computation can be conducted in R for the dyad-level SRM parameters, which in our example had a dyad-level sample size of $\sum D_g = N_d = 309$ observed interactions. Our OSF project (https://osf.io/2qs5w/) provides R syntax for these calculations at the dyad, case, and group levels.

2.2.3. Stage 2: Estimate SR-SEM Parameters

Estimation of the SR-SEM parameters described in Section 1.2.3 can proceed by passing Stage 1 estimated summary statistics as input data to standard SEM software. We used the R package `lavaan` [57] to separately specify and estimate a SEM for each level of analysis. It is also possible to specify a multilevel SR-SEM for multiple levels simultaneously by treating each level as an independent group in a multigroup SEM. We provide `lavaan` syntax to specify a multilevel SR-SEM as a multigroup SEM in our OSF project (https://osf.io/2qs5w/). Obtaining the true likelihood of the data would require constraints based on (average) network size, as suggested for ML-SEM estimation using "MuML" [63]. Determining the weights needed to apply such constraints lies beyond the scope of the current work and is unnecessary given the use of a residual-based rather than likelihood-ratio test statistic [10,11]) by minimizing the usual ML discrepancy function:

$$\hat{F}_{\text{ML}} = \log |\hat{\Sigma}_0| - \log |\hat{\Sigma}_1| + trace(\hat{\Sigma}_1 \hat{\Sigma}_0^{-1}) - p \\ + (\hat{\mu}_1 - \hat{\mu}_0)' \hat{\Sigma}_1^{-1} (\hat{\mu}_1 - \hat{\mu}_0), \tag{45}$$

where $p$ is the number of (components of) variables being modeled; $\hat{\Sigma}_1$ is the saturated-model Stage 1 estimate of a level-specific covariance matrix in Equations (6), (7), or (8) (analogous to a sample covariance matrix **S** of observed variables in standard SEM); and $\hat{\Sigma}_0 = \Sigma(\hat{\vartheta}_0)$ is a nested covariance matrix constrained as a function of estimated level-specific model parameters $\hat{\vartheta}$ (see Equations (28), (32), and (36) for population formulae). Analogous mean-structure parameters $\hat{\mu}_1$ and $\hat{\mu}_0 = \mu(\hat{\vartheta}_0)$ in the last added term of Equation (45) are only an option in the group-level model (Equation (28)). Note that the discrepancy function in Equation (45) is applied to any level-specific summary statistics, so it is not equivalent to the likelihood function in Ref. [8] (p. 877, Equation (19)), which is defined for observed round-robin variables rather than for their latent components.

The `lavaan` syntax to specify the structural model for the relations between the SRM components at the dyad level is presented in Appendix C. Syntax to specify SR-SEMs at the group and case levels can be found in our OSF project (https://osf.io/2qs5w/). In traditional SEM, there is no matrix of regression slopes among observed variables, only among latent variables (i.e., **B** in Equation (17)). However, when an observed variable $y$ is regressed on another in `lavaan` model syntax, `lavaan` implicitly "promotes" the observed variable to the latent space by treating it as a single-indicator factor (i.e., $\Lambda = \mathbf{I}$, so each $\lambda_{v,v} = 1$) without measurement error (i.e., $\Theta = \mathbf{0}$, so each $\theta_{v,v} = 0$). In `srm` syntax for FIML (also provided in our OSF project: https://osf.io/2qs5w/), the single-indicator factors must be specified explicitly to conduct a path analysis. In both `lavaan` and `srm` syntax, equality constraints representing indistinguishability can be specified by using the same label for multiple parameters (see Appendix C).

The $\hat{\Gamma}$ matrix can be passed to the `lavaan()` argument `NACOV=` in order to obtain corrected *SE*s for Stage 2 point estimates [10,11]; see Savalei [60] for more details. The

same $\hat{\Gamma}$ matrix is also used to calculate a residual-based test statistic [59] to test the null hypothesis of equal $\Sigma_0 = \Sigma(\vartheta)$ and saturated-model $\Sigma_1$ matrices. The end of Appendix C shows how to request $\chi^2$-based fit indices using the residual-based statistic, but we do not report fit indices here. Further research is warranted to investigate the sampling behavior of fit indices using this method. Other standard outputs are also available from `lavaan`, such as standardized solutions to report effect sizes and correlation residuals to assist diagnosing model misspecification.

## 3. Results

In this section, we report the Stage 1 and Stage 2 results for all three levels: dyad, case, and group. For Stage 1, we present the EAP results, which we use as input for Stage 2 estimation, although it is also possible to use other posterior summaries—such as the mode or maximum a posteriori (MAP) or the median (50th percentile) of the posterior distribution. In each of the following subsections, we provide two tables. The first presents Stage-1 point estimates: the (co)variances among the SRM components in the lower triangles (including the diagonal) and the corresponding correlations in the upper triangles (italicized). The second presents Stage-2 point and *SE* estimates, as well as standardized point estimates. We interpret Stage 2 results for pedagogical purposes, even if a parameter estimate is not statistically significant. We also compare our Stage 2 results to FIML [8] using the `srm` package, presented in the right columns of the second table in each subsection.

### 3.1. Dyad-Level SR-SEM Results

Dyad-level Stage 1 (co)variances and correlations are presented in Table 1. Given that the dyads were indistinguishable [13], covariances between the $ij$ and $ji$ components are constrained to equality (see Section 1.1.1)—for example, the *intra*personal covariances between preinteraction liking $ij$ and mimicry $ij$ (0.060) and the *inter*personal covariances between mimicry $ij$ and postinteraction liking $ji$ (0.102) in Table 1.

**Table 1.** Stage-1 covariance matrix among dyad-level SRM components.

| | SRM Component | 1 | 2 | 3 | 4 | 5 | 6 |
|---|---|---|---|---|---|---|---|
| 1. | Liking (pre) $R_{1,ij}$ | 0.750 | *.110* | *.090* | *.030* | *.346* | *.102* |
| 2. | Liking (pre) $R_{1,ji}$ | 0.083 | 0.750 | *.030* | *.090* | *.102* | *.346* |
| 3. | Social Mimicry $R_{2,ij}$ | 0.060 | 0.020 | 0.608 | *.657* | *.108* | *.147* |
| 4. | Social Mimicry $R_{2,ji}$ | 0.020 | 0.060 | 0.399 | 0.608 | *.147* | *.108* |
| 5. | Liking (post) $R_{3,ij}$ | 0.267 | 0.079 | 0.075 | 0.102 | 0.793 | *.229* |
| 6. | Liking (post) $R_{3,ji}$ | 0.079 | 0.267 | 0.102 | 0.075 | 0.182 | 0.793 |

*Note.* EAP estimates of covariances provided in the lower triangle (including the diagonal), EAP estimates of correlations (italicized) in the upper triangle.

Using the estimates in Table 1 as input data for Stage 2 resulted in a dyad-level structural model wherein $\beta^{(\mathbf{r})}_{61,inter} = \beta^{(\mathbf{r})}_{52,inter} = 0$. The residual-based $\chi^2(10) = 1.255$, $p > .999$, did not provide evidence against these constraints, so the hypothesis of exact data–model fit could not be rejected. However, the test statistic reported in the `summary()` output has too many $df$ because it does not account for equality constraints that follow from analyzing indistinguishable dyads. Even if we were to estimate the two slopes that were fixed to zero, we would still constrain those two slopes to equality ($\hat{\beta}^{(\mathbf{r})}_{61,inter} = \hat{\beta}^{(\mathbf{r})}_{52,inter}$), so our test statistic should really have only $df = 1$.

The problem is that the test statistic is derived by comparing the covariance matrix implied by estimates in Table 2 to estimates from a saturated model (in Table 1), in which equality constraints for indistinguishable dyads should also be specified. That is, there are only 12 unique parameter estimates in Table 1 (three variances, three dyadic reciprocities, three interpersonal covariances, and three intrapersonal covariances). Appendix C demonstrates how to appropriately specify a more constrained saturated model, whose $\chi^2 = 0$ but has $df = 9$ indistinguishability constraints (3 for variances, 3 for interpersonal

covariances, and 3 for intrapersonal covariances). The fitted model's test statistic is then the difference between the naïve statistics of the null-hypothesized model ($\chi^2_0$) and of the constrained saturated model ($\chi^2_1$):

$$\Delta\chi^2 = \chi^2_0 - \chi^2_1 = \chi^2_0 - 0 = \chi^2_0, \tag{46}$$

which is the same as the fitted model's $\chi^2_0$ because the saturated model fits perfectly ($\chi^2_1 = 0$). However, the correct degrees of freedom are calculated as the difference between the null-hypothesized model's $df_0$ and saturated model's $df_1$:

$$\Delta df = df_0 - df_1 = 10 - 9 = 1, \tag{47}$$

correctly reflecting the single (indistinguishable pair of) parameter(s) that were constrained to zero. Appendix C shows how to obtain the model's appropriate test statistic from `lavaan`, which remains not statistically significant, $\chi^2(1) = 1.255$, $p = .262$.

**Table 2.** Dyad-level structural parameters using two-stage MLE and FIML.

| Parameter | Two-Stage MLE | | | FIML | | |
|---|---|---|---|---|---|---|
| | Estimate | SE | Standardized | Estimate | SE | Standardized |
| | *Regression Slopes* | | | | | |
| $\beta^{(\mathbf{r})}_{31,intra}$ | 0.079 | 0.045 | 0.087 | 0.089 | 0.046 | 0.098 |
| $\beta^{(\mathbf{r})}_{53,intra}$ | −0.024 | 0.068 | −0.021 | −0.051 | 0.068 | −0.043 |
| $\beta^{(\mathbf{r})}_{51,intra}$ | 0.343 *** | 0.050 | 0.335 | 0.274 *** | 0.051 | 0.256 |
| $\beta^{(\mathbf{r})}_{41,inter}$ | 0.019 | 0.044 | 0.021 | 0.020 | 0.046 | 0.022 |
| $\beta^{(\mathbf{r})}_{63,inter}$ | 0.172 ** | 0.065 | 0.151 | 0.151 ** | 0.066 | 0.128 |
| | *(Residual) Covariances* | | | | | |
| $\psi^{(\mathbf{r})}_{21,dyadic}$ | 0.083 | 0.060 | 0.110 | 0.044 | 0.051 | .064 |
| $\psi^{(\mathbf{r})}_{43,dyadic}$ | 0.397 *** | 0.059 | 0.658 | 0.367 *** | 0.051 | .641 |
| $\psi^{(\mathbf{r})}_{65,dyadic}$ | 0.130 ** | 0.046 | 0.191 | 0.189 *** | 0.053 | .241 |
| | *(Residual) Variances* | | | | | |
| $\psi^{(\mathbf{r})}_{11}$ | 0.750 *** | 0.063 | 1.000 | 0.690 *** | 0.050 | 1.000 |
| $\psi^{(\mathbf{r})}_{33}$ | 0.602 *** | 0.059 | 0.992 | 0.567 *** | 0.051 | 0.990 |
| $\psi^{(\mathbf{r})}_{55}$ | 0.683 *** | 0.047 | 0.867 | 0.727 *** | 0.053 | 0.924 |

*Note.* Estimates significantly different from 0 flagged at two-tailed significance levels: ** $p < 0.01$, *** $p < 0.001$.

The parameters of this model, whose estimates are presented in Table 2, consist of intrapersonal and interpersonal regression slopes between SRM components of the three round-robin variables. For instance, the intrapersonal regression of postinteraction liking on preinteraction liking ($\beta^{(\mathbf{r})}_{51,intra}$) is interpreted as an autoregressive slope: controlling for person *i*'s mimicry of person *j* and person *j*'s mimicry of person *i* during a 5 min interaction (and given their person-level random effects), a 1-unit increase in person *i*'s preinteraction liking of person *j* is associated with a 0.343-unit average increase in person *i*'s postinteraction liking of person *j*. Likewise, the interpersonal regression of postinteraction liking on social mimicry ($\beta^{(\mathbf{r})}_{63,inter}$) is interpreted as follows: controlling for person *i*'s mimicry of person *j* during a 5 min interaction and person *i*'s preinteraction liking of person *j* (and given their person-level random effects), a 1-unit increase in person *j*'s social mimicry of person *i* during the 5 min interaction is associated with a 0.172-unit average increase in person *i*'s postinteraction liking of person *j*.

The dyadic covariance of preinteraction liking ($\psi^{(\mathbf{r})}_{21,dyadic}$) in Table 2 is the total covariance between the *ij* and *ji* components because they are exogenous. The standardized dyadic covariance is thus the dyadic reciprocity of preinteraction liking: person *i*'s liking of person *j* prior to their 5 min interaction was correlated $r = .110$ with person *j*'s liking

of person *i* prior to interacting (given their person-level random effects), which matches the correlation in Table 1. However, the dyadic covariances of endogenous variables social mimicry ($\psi^{(\mathbf{r})}_{43,dyadic}$) and postinteraction liking ($\psi^{(\mathbf{r})}_{65,dyadic}$) are residual covariances, so those standardized estimates are interpreted as partial correlations. Similarly, the relationship variance of preinteraction liking ($\psi^{(\mathbf{r})}_{11}$) is the total variance of the *ij* and *ji* components, whereas that of social mimicry ($\psi^{(\mathbf{r})}_{33}$) and postinteraction liking ($\psi^{(\mathbf{r})}_{55}$) are residual variances. Thus, the standardized residual variances of the endogenous SRM components are interpreted as the proportion of unexplained variance (i.e., $1 - R^2$). Only about 0.8% of variance in relationship-specific social mimicry was explained by their preinteraction liking, but preinteraction liking and social mimicry explained approximately 13% of relationship-specific postinteraction liking.

The parameter estimates of FIML and the two-stage estimator slightly differ from one another, perhaps due to the different MLE algorithms applied by `lavaan` and `srm`— i.e., `lavaan` maximizes the likelihood of observing a decomposed covariance matrix (for a specific level of analysis) estimated at Stage 1 as input data, whereas `srm` directly maximizes the likelihood of the raw round-robin data. However, both estimation approaches result in the same conclusions about the statistical significance of relations between dyad-level SRM components of the round-robin variables. Differences in dyad-level point and *SE* estimates were not systematically higher or lower using two-stage MLE rather than FIML.

### 3.2. Case-Level SR-SEM Results

Stage 1 results for the case level are presented in Table 3. In line with most published SRM results, the case-level (co)variances are smaller than those at the dyad level (compare to Table 1), indicating that much of the variability in peoples' liking and mimicry of others is quite dependent on their specific interaction partner. Consistent with the case-level SRM results of Salazar Kämpf et al. [13], differences in pre- and postinteraction liking were driven much more by alter (target) effects, whereas social mimicry was driven much more by ego (actor) effects.

**Table 3.** Stage 1 covariance matrix among case-level SRM components.

| | SRM Component | 1 | 2 | 3 | 4 | 5 | 6 |
|---|---|---|---|---|---|---|---|
| 1. | Liking (pre) $E_{1,i}$ | 0.056 | *.263* | *−.005* | *.117* | *.553* | *.178* |
| 2. | Liking (pre) $A_{1,i}$ | 0.031 | 0.240 | *.121* | *−.039* | *−.046* | *.721* |
| 3. | Social Mimicry $E_{2,i}$ | −0.001 | 0.035 | 0.346 | *.500* | *.178* | *.312* |
| 4. | Social Mimicry $A_{2,i}$ | 0.007 | −0.005 | 0.071 | 0.059 | *.203* | *.176* |
| 5. | Liking (post) $E_{3,i}$ | 0.026 | −0.004 | 0.021 | 0.010 | 0.038 | *−.029* |
| 6. | Liking (post) $A_{3,i}$ | 0.023 | 0.196 | 0.102 | 0.024 | −0.003 | 0.308 |

*Note.* EAP estimates of covariances provided in the lower triangle (including the diagonal), EAP estimates of correlations (italicized) in the upper triangle.

The case-level covariance matrix in Table 3 was used as input data to estimate SR-SEM parameters in Stage 2, using sample size $N = \sum N_g = 139$. In the case-level SR-SEM, we estimated the 10 unique ego–ego, alter–alter, ego–alter, and alter–ego regressions shown in Figure 2, as well as six (residual) variances and three (residual) covariances. Two regressions, $\beta^{(\mathbf{u})}_{61}$ and $\beta^{(\mathbf{u})}_{52}$, were constrained to zero. The residual-based $\chi^2(2) = 1.028$, $p = .598$ did not provide evidence against these constraints. Point and *SE* estimates are presented in Table 4, along with standardized point estimates. Standardized slopes are interpreted in units of each component's *SD*, standardized (residual) covariances are interpreted as (partial) correlations, and standardized residual variances are the proportion of each component's variance unexplained by its predictors (i.e., $1 - R^2$).

**Table 4.** Case-level structural parameters using two-stage MLE and FIML.

| Parameter | Two-Stage MLE | | | FIML | | |
|---|---|---|---|---|---|---|
| | **Estimate** | *SE* | **Standardized** | **Estimate** | *SE* | **Standardized** |
| *Regression Slopes* | | | | | | |
| $\beta_{31}^{(\mathbf{u})}$ | −0.099 | 0.487 | −0.040 | −0.067 | 0.128 | −0.133 |
| $\beta_{53}^{(\mathbf{u})}$ | 0.049 | 0.076 | 0.149 | 1.003 *** | 0.263 | 1.548 |
| $\beta_{51}^{(\mathbf{u})}$ | 0.450 | 0.275 | 0.545 | 0.123 | 0.073 | 0.375 |
| $\beta_{41}^{(\mathbf{u})}$ | 0.140 | 0.272 | 0.137 | 0.247 | 0.158 | 0.316 |
| $\beta_{54}^{(\mathbf{u})}$ | 0.053 | 0.221 | 0.065 | −0.217 | 0.129 | −0.516 |
| $\beta_{42}^{(\mathbf{u})}$ | −0.037 | 0.102 | −0.075 | −0.359 | 0.454 | −0.215 |
| $\beta_{64}^{(\mathbf{u})}$ | 0.272 | 0.390 | 0.120 | 0.958 *** | 0.163 | 0.910 |
| $\beta_{62}^{(\mathbf{u})}$ | 0.785 *** | 0.141 | 0.698 | 0.652 * | 0.264 | 0.370 |
| $\beta_{32}^{(\mathbf{u})}$ | 0.157 | 0.171 | 0.131 | 0.390 | 0.363 | 0.359 |
| $\beta_{63}^{(\mathbf{u})}$ | 0.160 | 0.119 | 0.171 | −0.191 | 0.305 | −0.118 |
| *(Residual) Covariances* | | | | | | |
| $\psi_{21}^{(\mathbf{u})}$ | 0.031 | 0.028 | 0.263 | 0.135 *** | 0.038 | .687 |
| $\psi_{43}^{(\mathbf{u})}$ | 0.073 | 0.043 | 0.519 | 0.071 ** | 0.031 | .426 |
| $\psi_{65}^{(\mathbf{u})}$ | −0.005 | 0.016 | −0.080 | −0.044 ** | 0.020 | −.387 |
| *(Residual) Variances* | | | | | | |
| $\psi_{11}^{(\mathbf{u})}$ | 0.056 | 0.040 | 1.000 | 0.421 *** | 0.068 | 1.000 |
| $\psi_{22}^{(\mathbf{u})}$ | 0.240 *** | 0.052 | 1.000 | 0.092 ** | 0.032 | 1.000 |
| $\psi_{33}^{(\mathbf{u})}$ | 0.340 *** | 0.071 | 0.984 | 0.099 ** | 0.035 | 0.919 |
| $\psi_{44}^{(\mathbf{u})}$ | 0.058 | 0.034 | 0.981 | 0.243 *** | 0.052 | 0.947 |
| $\psi_{55}^{(\mathbf{u})}$ | 0.025 | 0.017 | 0.660 | −0.056 * | 0.028 | −1.222 |
| $\psi_{66}^{(\mathbf{u})}$ | 0.129 ** | 0.040 | 0.426 | 0.037 | 0.035 | 0.129 |

*Note.* Estimates significantly different from 0 flagged at two-tailed significance levels: * $p < 0.05$, ** $p < 0.01$, *** $p < 0.001$.

These effects are interpreted similarly to linear regression effects, except that each regression coefficient quantifies the relation between two ego or alter components of the observed variables, not between the observed variables themselves. For example, the regression of the alter effect of postinteraction liking on the alter effect of preinteraction liking ($\beta_{62}^{(\mathbf{u})}$) is interpreted as an autoregressive slope: controlling for an actor *i*'s mimicry of others ($E_{2,gi}$) and others' mimicry of them ($A_{2,gi}$) during a 5 min interaction, a 1-unit increase in preinteraction liking of target *i* ($A_{1,gi}$) is associated with a 0.785-unit average increase in postinteraction liking of target *i* ($A_{3,gi}$). The regression of the alter effect of postinteraction liking on the ego effect of social mimicry ($\beta_{63}^{(\mathbf{u})}$) can be used to answer a research question about whether engaging in more social mimicry in general (across multiple partners) results in being liked more. The estimate $\hat{\beta}_{63}^{(\mathbf{u})} = 0.160$ implies that—given how much others already liked person *i* prior to interaction ($A_{1,gi}$) and how much they were mimicked by others ($A_{2,gi}$)—a 1-unit increase in person *i*'s mimicry of others ($E_{2,gi}$) is associated with a 0.160-unit average increase in how much others subsequently liked them ($A_{3,gi}$).

Similar to the dyad level, the generalized covariance between the ego and alter effects of preinteraction liking ($\psi_{21}^{(\mathbf{u})}$) is the total covariance between the ego and alter components of the variable, so its standardized estimate (a correlation) is generalized reciprocity. Likewise, the variance components of preinteraction liking ($\psi_{11}^{(\mathbf{u})}$ and $\psi_{22}^{(\mathbf{u})}$) are the total variances of the ego and alter components. The standardized $\hat{\Psi}^{*(\mathbf{u})}$ matrix provides partial correlations among residuals for endogenous variables social mimicry and postinteraction liking, which can be interpreted similarly to generalized reciprocity. For example,

$\psi_{43}^{*(\mathbf{u})} = .519$ indicates a strong partial correlation between person *i*'s mimicry of others and others' mimicry of them during a 5 min interaction, given preinteraction liking. As with dyad-level results, standardized residual variances indicate that little variance in social mimicry was explained by preinteraction liking (both $R^2 \approx 2\%$), whereas $R^2$ was quite high for both components of postinteraction liking ($R^2 = 34\%$ for ego effect, approximately 57% for alter effect).

The FIML results differ from those of two-stage MLE, at times leading to conflicting decisions regarding statistical significance and dissimilar standardized effect sizes. Sometimes the estimates were of opposite sign, but only for estimates that could not be statistically distinguished from zero. FIML yielded a significant negative residual variance for the ego component of postinteraction liking ($\psi_{55}^{(\mathbf{u})} = -0.056$), perhaps due to this parameter being nearly zero in the population. In fact, a normal theory CI around the positive two-stage estimate also includes small negative values.

### 3.3. Group-Level SR-SEM Results

Table 5 displays the group-level Stage 1 (co)variances and correlations of the dyadic variables, as well as the mean vector in the right column. Because liking was measured both before and after the 5 min interaction, it is interesting to note that liking seemed to increase after the interaction; however, the postinteraction composite included a third indicator (see Section 2.1), weakening the confidence in such a conclusion. As we expected, there was little group-level variance to extract from the variables (<1% of liking variance, 8% of mimicry variance) due to participants being randomly assigned to round-robin groups by Salazar Kämpf et al. [13]. In practice, it would be questionable whether group-level results should be interpreted, but we provide interpretations below for the pedagogical value of demonstrating the proposed SR-SEM.

**Table 5.** Stage 1 means and covariance matrix among group-level SRM components.

| | SRM Component | 1 | 2 | 3 | $\mu$ |
|---|---|---|---|---|---|
| 1. | Liking (pre) $\mu_{1,g}$ | 0.010 | .216 | .295 | 3.598 |
| 2. | Social mimicry $\mu_{2,g}$ | 0.007 | 0.087 | .329 | 2.968 |
| 3. | Liking (post) $\mu_{3,g}$ | 0.003 | 0.009 | 0.008 | 3.652 |

*Note.* EAP estimates of covariances provided in the lower triangle (including the diagonal), EAP estimates of correlations (italicized) in the upper triangle.

Stage 2 estimates are presented in Table 6. Estimates for FIML are unavailable, as `srm` does not provide group-level results. Note also that the model in Figure 3 is fully saturated, so no test of data–model fit is reported. We interpret regression slopes below, although none were significantly different from 0, which is no surprise given the small sample size at this level ($G = 26$ groups).

The group-level regression of postinteraction liking on preinteraction liking ($\beta_{31}^{(\mathbf{G})}$) indicates that when controlling for mimicry during a 5-min interaction, a 1-unit increase in average preinteraction liking among members of group *g* is generally associated with a 0.210-unit increase in their average postinteraction liking. Similarly, the group-level regression of postinteraction liking on social mimicry ($\beta_{32}^{(\mathbf{G})}$) is interpreted as follows: controlling for the average preinteraction liking within a group *g*, a 1-unit increase in average mimicry during a 5-min interaction leads to a 0.086-unit increase in average postinteraction liking within that group. As in dyad- and case-level results, standardized residual-variance estimates indicate that $R^2$ was higher for postinteraction liking (16.1%) than for social mimicry (4.7%).

**Table 6.** Group-level structural parameters using two-stage MLE.

| Parameter | Estimate | SE | Standardized |
|---|---|---|---|
| *Regression Slopes* | | | |
| $\beta_{21}^{(\mathbf{G})}$ | 0.623 | 1.180 | 0.216 |
| $\beta_{32}^{(\mathbf{G})}$ | 0.086 | 0.155 | 0.278 |
| $\beta_{31}^{(\mathbf{G})}$ | 0.210 | 0.493 | 0.235 |
| *Intercepts* | | | |
| $\nu_1$ | 3.598 *** | 0.065 | 35.152 |
| $\nu_2$ | 0.728 | 4.246 | 2.463 |
| $\nu_3$ | 2.644 | 1.859 | 29.004 |
| *(Residual) Variances* | | | |
| $\psi_{11}^{(\mathbf{G})}$ | 0.010 | 0.014 | 1.000 |
| $\psi_{22}^{(\mathbf{G})}$ | 0.083 | 0.067 | 0.953 |
| $\psi_{33}^{(\mathbf{G})}$ | 0.007 | 0.009 | 0.839 |

*Note.* Estimates significantly different from 0 flagged at two-tailed significance levels: *** $p < 0.001$.

## 4. Discussion

We proposed and demonstrated a two-stage estimation procedure for SR-SEM parameters. We adapted the method from similar proposals for estimating SEM parameters with multiply imputed data [10,11] and for estimating ML-SEM parameters from level-specific summary statistics [6]. We used the open-source software Stan [75] and R [72] packages `rstan` [74] and `lavaan` [57], with syntax provided in the Appendices and in our OSF project (https://osf.io/2qs5w/).

We also compared results with FIML estimation implemented in the `srm` package. Some differences were apparent, particularly at the case level, whose estimates are more variable due to less information (fewer people than dyads). The second author's master's thesis (available in a subdirectory of "Related preprints" in our OSF project: https://osf.io/2qs5w/) was a Monte Carlo simulation study that provided some preliminary evidence that neither estimator provides accurate case-level results for all parameters under these conditions. A reviewer also inquired about the possibility that results could differ because we modeled group-level effects, unlike the FIML estimator. In a subdirectory "noGroupsAnalysis" in our OSF project (https://osf.io/2qs5w/), we provide syntax files that exclude the group-level model at both stages, fitting those models to variables that were centered on the round-robin group means. Although results did show substantial differences in a few parameters (e.g., correlations differing greatly in magnitude, even switching signs), the parameters which were estimated with enough precision to be distinguishable from zero did not differ substantially. Furthermore, omitting the group-level model did not make Stage 2 results more similar to FIML. Nonetheless, this reveals the need for future simulation studies to investigate under what conditions the group-level model can influence results at other levels of analysis, despite those levels being orthogonal.

Below, we discuss some advantages of two-stage estimation over FIML, which derive from the flexibility of Stage 1 estimation and the availability of standard SEM software features in Stage 2. We also discuss some limitations of the two-stage approach and gaps in knowledge that need to be addressed before the two-stage approach can be applied with any confidence in the results.

### 4.1. Advantages and Limitations

Both single-stage FIML estimation and Stage 1 MCMC estimation become computationally intensive as the number of modeled variables increases, requiring greater estimation time than traditional least-squares estimators of round-robin effects. As the first SR-SEM developers demonstrated [8,9], the rewards for patience with greater computation time of FIML are less bias and greater CI coverage in small samples, as well as a simplified computational procedure to prepare fitting complex multivariate models. Regarding the limited-information SR-SEM estimator presented here, only Stage 1 MCMC estimation is

computationally intensive, whereas Stage 2 estimation is quite fast for any subsequent structural model(s) a researcher might be interested in fitting. In contrast, FIML would be computationally intensive for each structural model of interest, so a potential advantage of two-stage estimation is reduced computation time for researchers interested in fitting several SR-SEMs to the same data. An important next step for future simulation research will be to compare the quality of estimates provided by FIML and two-stage MLE.

In principle, numerous estimators could be used in Stage 1 to obtain summary statistics of SRM components. The limiting factor is whether one can calculate $\hat{\Gamma}$, which is necessary to adjust *SE*s and test statistics in Stage 2. If one is working with (approximately) normally distributed data, then single-stage FIML could be used for Stage 1 estimation by fitting a saturated SR-SEM via the `srm` package [71]), which also provides the ACOV in the `$vcov` element of the object returned by the `srm()` function. Although there might be little advantage to using a two-stage estimator when FIML is already available from the same software, one might be interested in fitting a SEM to only a single level of analysis. Whereas FIML requires fitting a model to both the case and dyad levels of analysis, we demonstrated that Stage 2 parameters can be estimated one level at a time (including the group level, if that is of interest). This means the fit of each level's model can be evaluated separately, without the complication of saturating the model of other levels of analysis, as has been recommended to evaluate ML-SEMs estimated with FIML [69]. Furthermore, estimating an unrestricted covariance matrix at each level in Stage 1 could prevent propagation of bias due to misspecification of either level's model in Stage 2—a potential advantage over single-stage FIML.

We used MCMC estimation of SRM summary statistics, which has some advantages over using MLE in Stage 1. The MLE available from `srm` assumes multivariate normality, without any robust corrections for non-normality that most standard SEM software packages provide [57,58]. Although our example Stan syntax (see Appendix A and our OSF project: https://osf.io/2qs5w/) only demonstrates using a multivariate normal likelihood, it is possible to instead specify a more appropriate distribution that captures the excess kurtosis that affects Type I error rates of normal theory test statistics in SEM [59]. For example, the Stan software has implemented a multivariate *t* distribution (find details in the *Stan Functions Reference*: https://mc-stan.org/docs/functions-reference/multivariate-student-t-distribution.html (accessed on 27 January 2024)), which can be used in place of a multivariate normal likelihood, so excess kurtosis can be modeled (or even estimated) via its degrees-of-freedom parameter. Doing so could yield more accurate estimates of the summary statistics' sampling variability, reflected by the posterior covariance matrix used to estimate $\Gamma$. Using $\hat{\Gamma}$ to adjust *SE*s and tests in Stage 2 could be sufficient to make them robust to non-normality, but future simulation research would be required to confirm this speculation.

In practice, round-robin variables might not be completely observed within a network. For example, a group member might not provide any responses (e.g., if a student was absent when data were collected from a classroom), preventing estimation of an ego effect. Yet responses could still be provided about the absent student, enabling estimation of their alter effect. Or within some dyads, there might only be information from one partner, due either to selecting a subset of peers who are friends [33] or to randomly assigning a subset of alters to each ego [32]. Incomplete data can be accommodated by the FIML estimator available in the `srm` package, but the MAR assumption would only be met when the SR-SEM includes all variables involved in the missing-data mechanism. Multiple imputation does not necessitate the analysis model becoming more complex because the imputation model in Stage 1 can incorporate such additional auxiliary variables. Similarly, the Stage 1 MCMC estimation we described could use data augmentation to accommodate incomplete data, functioning as an imputation model.

Two-stage estimation also facilitates the open-science practice of providing data with a published article for readers to reproduce analyses. Dyadic data have unique security concerns, given that participants could not only identify their own data but then use that

information to identify a partner's data [84]. Stage 2 SR-SEM parameters are estimated using only summary statistics and their ACOV, which can be shared without breaching the security of participant data.

A general advantage of MCMC estimation is the ability to incorporate existing knowledge by specifying informative prior distributions for parameters. Diffuse priors generally yield biased point estimates with small samples [85], and small round-robin groups are common in SRM applications (including our example data). Although we strove to specify weakly informative priors in Section 1.1.1, if they were too diffuse for these data, we could then expect inaccurate Stage 1 results to yield inaccurate Stage 2 estimates (garbage in, garbage out). We provide some preliminary simulation evidence in our OSF project (https://osf.io/2qs5w/) that confirmed this suspicion as well as revealed bias in case-level FIML estimates. Although the length of this paper prevents us from including an extensive simulation study to validate our proof of concept, we are currently investigating various methods of improving accuracy by incorporating empirical information into more "thoughtful priors" [86] or empirical Bayes priors. Despite introducing some statistical bias when the prior locations are inaccurate, informative priors also make estimates more precise, so biased estimates might be preferable if the overall mean-squared error is minimized [48,87]. Our goal is to reveal how the accuracy–precision trade-off [5] can be improved under small-sample conditions common in SRM research, and we will link the OSF project for this paper with those new OSF projects as they become available so readers will be able to find that new information.

Finally, the methods to correct *SE*s and test statistics are expected to work asymptotically, but in the case of multiple imputation, simulation studies have shown poorer performance in smaller samples [10,11]. Future simulation studies are needed to establish under which sample-size conditions (group size and number of groups) these methods can be expected to yield nominal Type I error rates in practice with round-robin data.

### 4.2. Extensions

The ML estimator available in the `srm` package does not accommodate case-level covariates, although Ref. [8] (p. 884, Footnote 2) mentions a "tedious procedure" to trick the software into estimating such case-level effects. Researchers are often interested in modeling case-level SRM components as predictors or outcomes [7,40], and even dyad-level covariates might be constant within a dyad rather than asymmetric [33,88]. The two-stage approach described here could be easily extended to accommodate level-specific covariates, including group-level variables, by expanding the level-specific correlation matrices and vectors of *SD*s to include such variables. Of course, this would require them to follow the same multivariate (normal or *t*) distribution as the SRM components.

For predictors that follow arbitrary distributions, their effects on round-robin components could instead be estimated in Stage 1 [7], yielding estimated summary statistics of their residuals. If `lavaan` were updated to accept a residual covariance matrix as data—along with exogenous-variable summary statistics and their estimated effects on endogenous variables—these could in principle be passed to `lavaan` as data, using the `conditional.x=TRUE` specification for fixed exogenous covariates [70].

Social and behavior scientists frequently measure variables using binary or ordinal (e.g., Likert) response scales, the latter of which can approximate a continuum with many (e.g., at least 5–7) response categories [89–91]. Having fewer response categories places limits on how large correlations can be, motivating the development of two-stage estimation that assumes that discrete observed responses are merely discretizations of latent normal responses [92], although that makes the normality assumption more difficult to test and impossible to correct for [93]. For ordinal or binary round-robin variables, the latent-response assumption could be applied during Stage 1 estimation of SRM summary statistics by incorporating a threshold model, similar to item factor analysis [94]. The summary statistics analyzed in Stage 2 would then be polychoric correlations and standardized thresholds [11].

## 5. Conclusions

Two-stage estimation of SR-SEM parameters is a promising development for researchers using round-robin data to answer research questions about complex interpersonal processes. Simulation studies are needed to validate its performance and establish best programming practices. Until such research is conducted, none of the details of the method presented here can be recommended as best practice; rather, this is merely a proof of concept. However, future research is justified by the flexibility of Stage 1 SRM estimation and the wealth of output available following Stage 2 SEM estimation.

**Author Contributions:** Conceptualization, T.D.J.; methodology, T.D.J. and Y.R.; software, T.D.J.; validation, T.D.J., A.M.B. and Y.R.; formal analysis, T.D.J. and A.M.B.; investigation, T.D.J. and A.M.B.; resources, T.D.J. and Y.R.; data curation, T.D.J. and A.M.B.; writing—original draft preparation, T.D.J. and A.M.B.; writing—review and editing, T.D.J., A.M.B. and Y.R.; visualization, T.D.J. and A.M.B.; supervision, T.D.J.; project administration, T.D.J.; funding acquisition, T.D.J. All authors have read and agreed to the published version of the manuscript.

**Funding:** This work was supported by the Dutch Research Council (NWO), project numbers 016.Veni.195.457 and 406.XS.01.078, awarded to Terrence D. Jorgensen.

**Institutional Review Board Statement:** Not applicable.

**Informed Consent Statement:** Not applicable.

**Data Availability Statement:** The publicly available data analyzed in this study are available from this Open Science Framework (OSF) project: https://osf.io/b4nvf/. We provide R and Stan syntax files to reproduce all of our reported analyses on our OSF project: https://osf.io/2qs5w/.

**Acknowledgments:** Many thanks to Salazar Kämpf and colleagues [13] for providing round-robin data via the OSF [12].

**Conflicts of Interest:** The authors declare no conflicts of interest. The funders had no role in the design of the study nor in the writing of the manuscript.

## Abbreviations

The following abbreviations are used in this manuscript:

| | |
|---|---|
| ACOV | Asymptotic (sampling) covariance matrix |
| AMEN | Additive and multiplicative effects model for network data |
| CI | Confidence interval |
| EAP | Expected a posteriori |
| (FI)ML(E) | (Full-information) maximum likelihood (estimation) |
| IRT | Item response theory |
| MAP | Maximum a posteriori |
| MCMC | Markov chain Monte Carlo |
| ML-SEM | Multilevel structural equation model |
| NACOV | $N$ times the asymptotic covariance matrix ($\Gamma$) |
| NUTS | No-U-Turn Sampler |
| OSF | Open Science Framework |
| PSRF | Potential scale-reduction factor |
| *SD* | Standard deviation |
| *SE* | Standard error |
| SEM | Structural equation model |
| SRM | Social relations model |
| SR-SEM | Social relations structural equation model |

## Appendix A. Stan Syntax Specifying a Multivariate Social Relations Model

```
data {
  // sample sizes
  int<lower=0> Nd;        // number of dyads   (Level 1)
  int<lower=0> Np;        // number of cases (Level 2, cross-classified)
  int<lower=0> Ng;        // number of groups  (Level 3)
```

```
                   // number of observed measures (half the number of columns)
                   int<lower=0> Kd2;        // number of round-robin variables
                   // observed data
                   matrix[Nd, 2*Kd2] Yd2;  // observed round-robin variables
                   // ID variables in dyad-level data set
                   int IDp[Nd, 2];          //  case-level IDs (cross-classified)
                   int IDg[Nd];             // group-level IDs
               }

               parameters {
                 // means
                 vector[Kd2] Mvec;                    // round-robin variables
                 // SDs
                 vector<lower=0>[ Kd2 ]  s_rr;  // round-robin residuals
                 vector<lower=0>[2*Kd2] S_p;     //  case-level (AP) effects
                 vector<lower=0>[ Kd2 ] S_g;     // group-level (GG) effects

                 // Cholesky factor of correlation matrices for random effects
                 cholesky_factor_corr[ Kd2 ] chol_g; // group-level (GG)
                 cholesky_factor_corr[2*Kd2] chol_p; //  case-level (AP)

                 // correlations among round-robin residuals
                 //  - dyadic reciprocity (within variable) on diagonal
                 //  - intrapersonal correlations (between variable, within  case) below diagonal
                 //  - interpersonal correlations (between variable, between case) above diagonal
                 matrix<lower=0,upper=1>[Kd2, Kd2] r_d2;

                 // random effects to sample on unit scale
                 matrix[Np, 2*Kd2] AP; // matrix of all ego and all alter effects
                 matrix[Ng,   Kd2] GG; // group-level random effects
               }

               transformed parameters {
                 // expected values, given random effects
                 matrix[Nd, 2*Kd2] Yd2hat;   // dyad-level \hat{y}s
                 // combined dyad-level SDs and correlations
                 vector[2*Kd2] S_d;
                 matrix[2*Kd2, 2*Kd2] Rd2;
                 // cholesky decomposition of dyad-level correlation matrix
                 matrix[2*Kd2, 2*Kd2] chol_d;

                 // combine correlations among round-robin variables
                 {
                   int idx1;  // arbitrary iterators
                   int idx2;
                   int idp1;
                   int idp2;

                   for (k in 1:Kd2) {
                      idx1 = k*2 - 1;
                      idx2 = k*2;

                      // within round-robin variable
                      S_d[idx1] = s_rr[k];          // equal relationship SDs
                      S_d[idx2] = s_rr[k];
                      Rd2[idx1, idx1] = 1;          // diagonal = 1
                      Rd2[idx2, idx2] = 1;
                      Rd2[idx1, idx2] = -1 + 2*r_d2[k,k];  // equal dyadic reciprocity
                      Rd2[idx2, idx1] = -1 + 2*r_d2[k,k];

                      // between round-robin variables
                      if (k < Kd2) { for (kk in (k+1):Kd2) {
                        idp1 = kk*2 - 1;
                        idp2 = kk*2;

                        Rd2[idx1, idp1] = -1 + 2*r_d2[kk, k ]; // within case (intra = BELOW)
                        Rd2[idx2, idp2] = -1 + 2*r_d2[kk, k ];
                        Rd2[idp1, idx1] = -1 + 2*r_d2[kk, k ];
                        Rd2[idp2, idx2] = -1 + 2*r_d2[kk, k ];
                        Rd2[idx1, idp2] = -1 + 2*r_d2[k , kk]; // between case (inter = ABOVE)
                        Rd2[idp2, idx1] = -1 + 2*r_d2[k , kk];
                        Rd2[idx2, idp1] = -1 + 2*r_d2[k , kk];
                        Rd2[idp1, idx2] = -1 + 2*r_d2[k , kk];
                      }}

                   }

                   // end block combining dyad-level correlation matrix
                 }
                 // cholesky decompositions for model{} block
                 chol_d = diag_pre_multiply(S_d, cholesky_decompose(Rd2));
```

```
// calculate and/or combine expected values
{
  int idx1;  // arbitrary iterators
  int idx2;

  for (k in 1:Kd2) {
    idx1 = k*2 - 1; // ego effect for k^th measure
    idx2 = k*2;   // alter effect for k^th measure

    for (d in 1:Nd) {
      // expected values of round-robin variables , given random effects
      Yd2hat[d, idx1] = Mvec[k] + (S_g[k]*GG[ IDg[d], k]) +
            S_p[idx1]*AP[ IDp[d,1], idx1] + S_p[idx2]*AP[ IDp[d,2], idx2];
      Yd2hat[d, idx2] = Mvec[k] + (S_g[k]*GG[ IDg[d], k]) +
            S_p[idx1]*AP[ IDp[d,2], idx1] + S_p[idx2]*AP[ IDp[d,1], idx2];
    }

  }
}

}

model {
  // priors for means and SDs , based on empirical ranges
  for (k in 1:Kd2) {
    // means
    Mvec[k]   ~ normal(3.5, 1); // 1-6 Likert scale , variances approx. 1
    // residual/dyadic SDs
    s_rr[k]       ~ student_t(4, 0.50, 0.5) T[0, ];
    // ego effect SDs
    S_p[2*k - 1] ~ student_t(4, 0.25, 0.5) T[0, ];
    // alter effect SDs
    S_p[2*k]      ~ student_t(4, 0.25, 0.5) T[0, ];
    // group effect SDs
    S_g[k]         ~ student_t(4, 0.05, 0.5) T[0, ];
  }

  // priors for correlations
  chol_g ~ lkj_corr_cholesky(2);
  chol_p ~ lkj_corr_cholesky(2);
  for (k in 1:Kd2) {
    // dyadic correlations (priors on diagonal)
    r_d2[k,k] ~ beta(1.5, 1.5);
    // between-variable correlations
    if (k < Kd2) { for (kk in (k+1):Kd2) {
      r_d2[kk, k ] ~ beta(1.5, 1.5); // intra = BELOW
      r_d2[k , kk] ~ beta(1.5, 1.5); // inter = ABOVE
    }}
  }

  // priors for random effects
  for (n in 1:Np) AP[n,] ~ multi_normal_cholesky(rep_row_vector(0, 2*Kd2), chol_p);
  for (n in 1:Ng) GG[n,] ~ multi_normal_cholesky(rep_row_vector(0,   Kd2), chol_g);

  // likelihoods for observed data
  for (n in 1:Nd) Yd2[n,] ~ multi_normal_cholesky(Yd2hat[n,], chol_d);
}

generated quantities{
  matrix[   Nd, 2*Kd2] Yd2e; // residuals (relationship effects + error)
  matrix[2*Kd2, 2*Kd2] Rp;   // case-level correlation matrix
  matrix[ Kd2 ,   Kd2 ] Rg;   //  group-level correlation matrix
  // % variance at group, case, and dyad levels
  matrix[Kd2, 4] Rsq;

  // calculate % of each round-robin variance due to each random effect
  {
    matrix[Kd2, 4] vars; // group, ego, alter, and relationship variances
    vector[Kd2] totals;  // sum of variance components

    for (k in 1:Kd2) {
      vars[k,1] = square(S_g[k]);        // group
      vars[k,2] = square(S_p[2*k - 1]); // ego
      vars[k,3] = square(S_p[2*k]);     // alter
      vars[k,4] = square(s_rr[k]);       // relationship

      totals[k] = sum(vars[k, ]);
      Rsq[k,] = vars[k,] ./ totals[k];
    }
    // end R-squared block
```

```
  }

  // calculate residuals to return as relationship effects
  Yd2e = Yd2 - Yd2hat;

  // calculate case-level correlation matrix
  Rp = multiply_lower_tri_self_transpose(chol_p);
  // calculate group-level correlation matrix
  Rg = multiply_lower_tri_self_transpose(chol_g);
}
```

## Appendix B. R Syntax to Fit Multivariate Social Relations Model to Data

```
library(rstan)
rstan_options(auto_write = TRUE)

## known quantities to pass to Stan's data{} block
knowns <- list(Nd = nrow(dat),                # number of dyads   (Level 1)
               Np = length(caseIDs),          # number of cases   (Level 2)
               Ng = length(unique(dat$Group)), # number of groups  (Level 3)
               Kd2 = 3, # number of round-robin variables
               ## round-robin data in wide format
               ## (1 row per dyad, 2 columns per round-robin variable)
               Yd2 = as.matrix(dat[,c("preLike_ij","preLike_ji",
                                       "mimicry_ij","mimicry_ji",
                                       "postLike_ij","postLike_ji")]),
               ## case-level IDs per dyad
               IDp = as.matrix(dat[,c("ID_i","ID_j")]),
               IDg = array(dat$Group)) # group ID per dyad

## unknown quantities to be sampled in Stan's parameters{} block, or saved
## from Stan's transformed parameters{} or generated quantities{} blocks.

mu <- "Mvec"                           # mean vector of round-robin variables
sigma <- c("s_rr","S_p","S_g")         # SDs of round-robin variable components
corr <- c("Rd2","Rp","Rg")             # correlations among round-robin components
derived <- c("Rsq")  # proportion of variance account for by each component
unknowns <- c(mu, sigma, corr, derived)

## fit model

stage1 <- stan(file = "AppendixA.stan", data = knowns, pars = unknowns,
               seed = 3141593, chains = 4, cores = 4,
               iter = 3000, ## default warmup = iter/2 = 1500 in this case
               init_r = .5) # to prevent nonpositive definite starting values
```

*Appendix B.1. R Syntax to Prepare Stage 1 Dyad-Level Results for Stage 2 Analysis*

```
## round-robin variable names
vn <- c("preLike","mimicry","postLike")
## names of dyad-level components
vnD <- paste(rep(vn, each = 2), c("ij","ji"), sep = "_")

## stack posterior samples of estimated summary statistics
RdVec <- do.call(rbind, As.mcmc.list(stage1, pars = "Rd2"))
# head(RdVec)
SdVec <- do.call(rbind, As.mcmc.list(stage1, pars = "s_rr"))
# head(SdVec)

## store correlations in a matrix (per posterior sample)
RdList <- apply(RdVec, MARGIN = 1, FUN = function(x) {
  Rd2 <- matrix(0, nrow = length(vnD), ncol = length(vnD),
                dimnames = list(vnD, vnD))
  for (i in names(x)) eval(parse(text = paste(i, "<-", x[i]) ))
  Rd2
}, simplify = FALSE)
# RdList[1:2]

## store SDs in a diagonal matrix (per posterior sample)
SdList <- apply(SdVec, MARGIN = 1, FUN = function(x) {
  s_rr <- numeric(length(vnD)/2) # equality constraints
  for (i in names(x)) eval(parse(text = paste(i, "<-", x[i]) ))
  Sd <- diag(rep(s_rr, each = 2)) # repeat equal variances
  dimnames(Sd) <- list(vnD, vnD)
  Sd
}, simplify = FALSE)
# SdList[1:2]

## scale correlations to covariance matrices
```

```
dSigmaList <- mapply(function(R, S) S %*% R %*% S,
                     R = RdList, S = SdList, SIMPLIFY = FALSE)
# dSigmaList[1:2]

## posterior mean of estimated (co)variances
SigmaD <- Reduce("+", dSigmaList) / length(dSigmaList)
class(SigmaD) <- c("lavaan.matrix.symmetric","matrix")
SigmaD
cov2cor(SigmaD) # posterior mean of estimated correlations

## vectorize all (co)variances per posterior sample
ACOV.list.D <- lapply(dSigmaList, function(S) {
  ## lavaan expects this order:
  lavaan::lav_matrix_vech(S, diagonal = TRUE) # lower.tri covariances
})
# ACOV.list.D[[1]]
## posterior (co)variability of all (co)variances,
## multiplied by 309 subjects (i.e., an estimate of the "Gamma" matrix)
NACOV.d <- 309 * cov(do.call(rbind, ACOV.list.D))
# check dimensions:  nrow(NACOV.d) == length(vnD) * (length(vnD) + 1L) / 2L
```

## Appendix C. R Syntax to Fit the Social Relations Structural Equation Model

```
library(lavaan)

## specify dyad-level path model

path.d <- '
  ## Intra Effects (ij)
    mimicry_ij   ~ intra_a*preLike_ij
    postLike_ij ~ intra_b*mimicry_ij + intra_c*preLike_ij
  ## Intra Effects (ji)
    mimicry_ji   ~ intra_a*preLike_ji
    postLike_ji ~ intra_b*mimicry_ji + intra_c*preLike_ji
  ## i-to-j Effects
    mimicry_ji   ~ inter_a*preLike_ij
    postLike_ji ~ inter_b*mimicry_ij
  ## j-to-i Effects
    mimicry_ij   ~ inter_a*preLike_ji
    postLike_ij ~ inter_b*mimicry_ji

  ## (residual) covariances
    preLike_ij   ~~ preLike_ji
    mimicry_ij   ~~ mimicry_ji
    postLike_ij ~~ postLike_ji

  ## equal variances
    preLike_ij   ~~ var_x*preLike_ij
    preLike_ji   ~~ var_x*preLike_ji
    mimicry_ij   ~~ var_m*mimicry_ij
    mimicry_ji   ~~ var_m*mimicry_ji
    postLike_ij ~~ var_y*postLike_ij
    postLike_ji ~~ var_y*postLike_ji

  ## hypothesized indirect effect
    ind := intra_a*inter_b
'
fit.d <- sem(path.d, sample.cov = SigmaD, sample.nobs = 309,
             h1 = FALSE, baseline = FALSE,
             NACOV = NACOV.d, sample.cov.rescale = FALSE, fixed.x = FALSE,
             se = "robust.sem", test = "Browne.residual.adf")

## CANNOT TRUST model-fit test statistics in summary() output (wrong df)

## to test model fit, specify a "saturated" model with equality constraints
## reflecting round-robin design
sat.mod.d <- ' ## equal variances
    preLike_ij   ~~ var_x*preLike_ij
    preLike_ji   ~~ var_x*preLike_ji
    mimicry_ij   ~~ var_m*mimicry_ij
    mimicry_ji   ~~ var_m*mimicry_ji
    postLike_ij ~~ var_y*postLike_ij
    postLike_ji ~~ var_y*postLike_ji
  ## dyadic reciprocity
    preLike_ij   ~~ rec_x*preLike_ji
    mimicry_ij   ~~ rec_m*mimicry_ji
    postLike_ij ~~ rec_y*postLike_ji
  ## equal intrapersonal covariances
    preLike_ij ~~ intra_xm*mimicry_ij  + intra_xy*postLike_ij
    mimicry_ij ~~ intra_my*postLike_ij
```

```
      preLike_ji ~~ intra_xm*mimicry_ji  + intra_xy*postLike_ji
      mimicry_ji ~~ intra_my*postLike_ji
   ## equal interpersonal covariances
      preLike_ij ~~ inter_xm*mimicry_ji  + inter_xy*postLike_ji
      mimicry_ij ~~ inter_my*postLike_ji
      preLike_ji ~~ inter_xm*mimicry_ij  + inter_xy*postLike_ij
      mimicry_ji ~~ inter_my*postLike_ij
'
sat.fit.d <- lavaan(sat.mod.d, sample.cov = SigmaD, sample.nobs = 309,
                    h1 = FALSE, baseline = FALSE,
                    NACOV = NACOV.d, sample.cov.rescale = FALSE,
                    se = "robust.sem", test = "Browne.residual.adf")
## store model inside lavaan object, where they will override the default
## saturated model when running lavTestLRT()
fit.d@external$h1 <- sat.fit.d

## Now lavTestLRT() calculates the residual-based test with correct df and p value
lavTestLRT(fit.d, type = "Browne.residual.adf")

## The tests in the summary() still have the WRONG degrees of freedom.
## ONLY interpret parameter estimates (and their test statistics).
summary(fit.d, std = TRUE, rsquare = TRUE)
```

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
