# Peer review of "Two-Stage Limited-Information Estimation for Structural Equation Models of Round-Robin Variables"

_stats, doi:10.3390/stats7010015_

Round 1

Reviewer 1 Report

Comments and Suggestions for Authors

Round Robin data or social network data is used in a variety of disciplines to answer a wide range of substantive research questions. However, the analysis of the (dyadic) data is complex, because it is characterized by complex dependencies that must be considered in the statistical model. In the submitted article, a two-stage structural equation modeling (SEM) approach is proposed in which first (1) the covariance matrices of the group, person and dyad effects are determined. These matrices are then (2) used for further analysis in standard SEM software to estimate the relationships of interest. The approach is illustrated with an example data set and open questions for the future are discussed at the end of the manuscript.

I read the article with great interest and think it makes an interesting and important contribution to round-robin data modeling. I have only a few minor comments that should be easily addressed by the authors.

1.  The first part of the article (section 1) is very dense. This is to be expected given the topic, but I would have found it useful if the authors had already briefly explained at the end of the introduction (pages 1-2) what the basic idea of their approach is and how the subsequent parts fit in with the proposed approach. This would make it clearer why reading the following sections is “important”.

2. On page 2, lines 44-46, the authors write that they will list limitations of the FIML approach in section 1.2.3. However, they do not actually do this, but only - if at all - in the discussion section (the mention in section 1.2.3. that the SR-SEM approach in Nestler et al. does not contain a model for the group-level structure. This is true, but it is certainly not a limitation).

3. Citation 5 and 6 are identical.

4. I am not an expert on Stan, so this may be the reason why I did not understand how the authors manage to keep the covariance matrix of the dyad effects positive definite. In WinBugs one would have to explicitly define constraints and discard the draw if necessary. Is this automatically achieved in Stan by the choice of the priors? Or is it automatically implemented in the sampling process? Perhaps making this more explicit would help here.

5. The authors probably chose Stan for a reason, but is it possible to provide JAGS code to estimate the sufficient statistics in the first step?

6. I found it surprising that the results of FIML and the 2-stage approach differ for the person effects. One possible explanation might be that the authors include a model at the group level (although the group effects should be orthogonal to the person effects). I was wondering whether the results are more similar, when the authors do not include the group structure in their model.

7. Another advantage of the authors' approach is that it more clearly separates the level of the individuals from the level of the dyad in the analysis. By first estimating saturated covariance matrices per level, model misspecifications at the other level should not play a role, right? This should, for example, make it easier to assess the fit at the different levels.

Comments on the Quality of English Language

I have no comments with regard to this.

Author Response

Thank you for your positive feedback and constructive criticism.  The list below describes how we addressed each of your listed concerns.

  1. Thank you for the suggestion.  We agree the introduction could be improved, and we hope you find the revision more suitable. We now begin by discussing the general utility of two-stage estimation in latent-variable modeling, providing common examples for context. We mention the particular methods that inspired our proposal, discussing more thoroughly how the paper is structured to provide details about the background material.
  2. This phrase no longer appears in the revised introductory paragraphs.
  3. Citation 5 (now 12) is for the OSF project that hosts the data we analyzed, whereas Citation 6 (now 13) is for the paper that reports how the data were collected and originally analyzed. We hope our text is sufficiently clear in our revised introduction: "...empirical data made publicly available on the Open Science Framework [OSF; 12] by Salazar Kämpf et al. [13]."
  4. Indeed, our first version hinted at the solution, but without providing sufficient information.  The blue text in our revised Section 2.2.1 now provides more practical information about how Stan maintains positive definiteness by "learning" the problematic areas of the parameter space during adaptation.  This is the same solution that the blavaan package relies on, although their paper does not provide sufficient details about this when they described their priors for correlations. I (the first author) had to learn about it from a conversation with the blavaan maintainer.  So we are glad to have it stated more explicitly in this paper, although the technical details about Stan's adaption phase are still omitted (which we think are quite beyond the scope of the present article).
  5. Related to the point above, we chose Stan precisely because it did not require the complex dyad-level correlation constraints that would be necessary using BUGS/JAGS.  The blue text in our revised Section 2.2.1 now mentions the lack of feasibility of a BUGS implementation as a motivating factor for using Stan's NUTS algorithm.
  6. Thank you for the suggestion. We agree that the orthogonality of group-level effects did not lead us to expect differences.  Interestingly, there were substantial differences in the case-level results, although mainly among parameters that were already estimated without much precision.  There was no easily discernible pattern to the differences, nor did the results become systematically more similar to FIML results.  The second paragraph of the Discussion now mentions all this.
  7. Thank you for suggesting this additional advantage.  We have now added this to the end of the second paragraph of Section 4.1, immediately after mentioning the advantage of separately evaluating the fit of each level's model: "Furthermore, estimating an unrestricted covariance matrix at each level in Stage 1 could prevent propagation of bias due to misspecification of either level’s model in Stage 2—a potential advantage over single-stage FIML."

Any revised text can be found in blue font.  If blue is not a legible color for the reviewer(s), we are happy to change it to another color of their choice.

Reviewer 2 Report

Comments and Suggestions for Authors

1. Rewrite the abstract of the paper so that it includes the purpose of the research paper, the methods used, and the results reached by the paper. The abstract included mentioning some references, which is incorrect in my opinion.

2. The introduction is very long, and it is better to shorten it to the extent that it achieves its purpose.

3.The conclusions do not reflect the research results that the author believes he has reached.

4.There are some added paragraphs in the paper that are unnecessary, in my opinion, such as extensions on page 24 and notes on page 29.

Comments on the Quality of English Language

Moderate editing of English language required

Author Response

  1. The abstract has been rewritten to reflect "the purpose of the research paper, the methods used, and the results reached by the paper", in that order. The revised abstract does not include references.
  2. The introduction covers the background necessary for readers to understand the proposed method and how it fits within the larger literature of two-stage estimation of SEM parameters. The other reviewers do not share the opinion that it is too long, but one reviewer did recommend that our introduction provide a better explanation of how it was organized.  The first 2 paragraphs of the revised introduction now provide the justification for the components of the introduction.
  3. This comment is not specific enough to permit a response. Perhaps the reviewer shares the opinion of another reviewer, who requested a simulation to shed light on differences between the proposed estimator and FIML.  Our revision now addresses that.
  4. The reviewer is welcome to their opinion.  It is common to discuss future directions of research at the end of a paper. It is particularly appropriate in this case because we have provided a proof-of-concept whose biggest advantages over FIML would lie in the extensions we discuss.  Regarding the Endnotes, we agree that these useful bits of additional information are digressions, which is why they are Endnotes rather than appearing in the main text. 

Reviewer 3 Report

Comments and Suggestions for Authors

  1. In line 467-476, a LKJ distribution with eta = 2 is specified as the prior for the group and case level correlation matrices. Given the range of correlation is between -1 and 1, it will be helpful to clarify if “higher” and “smaller” correlations are compared in absolute value.

  2. In section 2.2.1, the variance in the prior distribution of mu_G, group-level variance, person-level effects and dyadic effects are small. These distributions either follow a normal distribution with small variance or a t distribution with degrees of freedom equal to 4 and a scale parameter sigma of 0.5. This choice appears to be informative, as indicated in line 465, where it is mentioned that most prior probability mass falls between 0 and 1.5. However, the selection of 1.5 seems somewhat arbitrary. Typically, an inverse gamma distribution is conventionally employed for the prior distribution of variance, so it raises questions about why t distributions are opted for in this context. Clarification on this choice would enhance understanding.

  3. In line 515-521, it is noted that the Gelman-Rubin statistic was used to check for convergence which was assumed once the potential scale reduction factor became less than 1.05. However, in many cases, this can occur long before convergence is  achieved. Also, the PSRF is a univariate diagnostic, but given the multiple parameters in the model, have the authors checked the multivariate diagnostic such as multivariate potential scale reduction factor described by Brooks and Gelman (1998)?

  4. By using the real world data, the authors compared estimates of multiple parameters by using the proposed two-stage MLE and FIML models. While this empirical examination offers valuable insights, it would be beneficial for the authors to supplement their findings with simulation studies. Specifically, extending the comparison to include simulated scenarios at both dyad and case levels would provide a more comprehensive understanding of the performance of the two models under varying conditions.

Author Response

Thank you for your suggestions, which we address in order below:

  1. Indeed, we meant smaller in absolute value.  We added blue text in that paragraph (below Eq. 41) to clarify this.
  2. As you suggested, we have added some blue text in Section 2.2.1 to more carefully explain our choice of prior hyperparameters regarding the mean vector and random-effect SDs, citing Gelman's (2006) paper proposing the use of the half-t family for scale parameters (and demonstrating why it should be preferred over inverse-Gamma priors on variances), as well as a simulation study investigating its use in a variance-decomposition model.  As we now more carefully explain, we strove to choose "weakly informative" prior scales, which would minimally influence the posterior.  We do not specify arbitrarily large variances, as is common in the Jeffreys tradition and BUGS programmers (e.g., the Mplus defaults have effectively "infinite" variances) because they place too much information in parts of the parameter space that are unreasonable for the data.  But although our prior-SDs are smaller, they should not be considered "small" or very informative, but instead are only weakly informative, given the scale of the observed variables that are being decomposed into 4 variance components.  
  3. We agree, and we added text (in blue font) to the end of Section 2.2.1 indicating that we inspected convergence using traceplots, that verified adequate mixing of chains.  We increased the number of iterations to 5000 per chain (discarding half as burn-in), and we now report the range of effective sample sizes (min = 282), as well as the maximum univariate PSRF (1.01) and the multivariate PSRF (1.03). All statistics reported in the Results (including Tables 1-6) have been updated accordingly.
  4. We agree with the need for evidence from Monte Carlo simulation studies to validate the proposed method and shed light on differences with the FIML estimator.  We have several such studies in various stages of progress, one of which was a master-thesis project completed by the second author (now uploaded to our OSF project), and the other is an in-press chapter of the 2023 IMPS proceedings that investigates Stage-1 SRM estimates (also uploaded to our OSF project).  In addition to our manuscript already being quite long, we decided to omit a simulation study from it because our preliminary studies were not sufficiently conclusive.  The thesis, for example, simulated data based on the parameter estimates reported in this manuscript, which include some rather small variance components.  This boundary condition yielded some strange patterns of bias in both the FIML and 2-stage estimates, although Nestler et al.'s (2020, 2022) simulations did not report substantial bias.  Assuming this was due to the boundary condition, we opted to instead simulate data from Nestler et al.'s (2020) population in our IMPS chapter, which still revealed substantial underestimation of many case-level parameters using FIML, and even more bias of MCMC estimates. We tentatively concluded that the priors were too diffuse (not sufficiently informative) for small groups, which led to our current project about comparing various methods of specifying "thoughtful" priors based on available empirical information.  These simulation studies will be reported in separate papers because there is not enough room in the current proof-of-concept paper to sufficiently represent the variety of contexts and associated caveats for best practices.  The penultimate paragraph of our Discussion already vaguely hinted at some of these issues and our ongoing research, but our revised paragraph (in blue font) points readers to our evidence on OSF about biased case-level estimates, which indicate that neither the FIML nor 2-stage can be assumed to be very accurate.  We will update our OSF project with links to new simulation studies as they become available.

Any revised text can be found in blue font.  If blue is not a legible color for the reviewer(s), we are happy to change it to another color of their choice.

Round 2

Reviewer 3 Report

Comments and Suggestions for Authors

I think that the authors have adequately addressed the comments made by the reviewers in the revised version of the manuscript. Therefore, I have no further comments.